# MODEL-BASED MICRO-DATA REINFORCEMENT LEARNING: WHAT ARE THE CRUCIAL MODEL PROPERTIES AND WHICH MODEL TO CHOOSE?

**Balázs Kégl, Gabriel Hurtado, Albert Thomas**
Huawei Noah's Ark Lab, Paris, France
`{balazs.kegl,gabriel.hurtado,albert.thomas}@huawei.com`

## ABSTRACT

We contribute to micro-data model-based reinforcement learning (MBRL) by rigorously comparing popular generative models using a fixed (random shooting) control agent. We find that on an environment that requires multimodal posterior predictives, mixture density nets outperform all other models by a large margin. When multimodality is not required, our surprising finding is that we do not need probabilistic posterior predictives: deterministic models are on par, in fact they consistently (although non-significantly) outperform their probabilistic counterparts. We also found that heteroscedasticity at training time, perhaps acting as a regularizer, improves predictions at longer horizons. At the methodological side, we design metrics and an experimental protocol which can be used to evaluate the various models, predicting their asymptotic performance when using them on the control problem. Using this framework, we improve the state-of-the-art sample complexity of MBRL on Acrobot by two to four folds, using an aggressive training schedule which is outside of the hyperparameter interval usually considered.

## 1 INTRODUCTION

Unlike computers, physical systems do not get faster with time (Chatzilygeroudis et al., 2020). This is arguably one of the main reasons why recent beautiful advances in deep reinforcement learning (RL) (Silver et al., 2018; Vinyals et al., 2019; Badia et al., 2020) stay mostly in the realm of simulated worlds and do not immediately translate to practical success in the real world. **Our long term research agenda is to bring RL to controlling real engineering systems.** Our effort is hindered by slow data generation and rigorously controlled access to the systems.

Micro-data RL is the term for using RL on systems where the main bottleneck or source of cost is access to data (as opposed to, for example, computational power). The term was introduced in robotics research (Mouret, 2016; Chatzilygeroudis et al., 2020). This regime requires performance metrics that put as much **emphasis on sample complexity** (learning speed with respect to sample size) as on asymptotic performance, and algorithms that are designed to make efficient use of small data. Engineering systems are both tightly controlled for safety and security reasons, and physical by nature (so do not get faster with time), making them a primary target of micro-data RL. At the same time, engineering systems are the backbone of today's industrial world: controlling them better may lead to multi-billion dollar savings per year, even if we only consider energy efficiency.[1]

Model-based RL (MBRL) builds predictive models of the system based on historical data (logs, trajectories) referred to here as *traces*. Besides improving the sample complexity of model-free RL by orders of magnitude (Chua et al., 2018), these models can also contribute to adoption from the human side: system engineers can "play" with the models (data-driven generic "neural" simulators) and build trust gradually instead of having to adopt a black-box control algorithm at once (Argenson & Dulac-Arnold, 2020). **Engineering systems suit MBRL particularly well in the sense that most system variables that are measured and logged are relevant**, either to be fed to classical control or to a human operator. This means that, as opposed to games in which only a few variables (pixels) are relevant for winning, learning a forecasting model in engineering systems for the *full* set of logged variables is arguably an efficient use of predictive power. It also combines well with the micro-data learning principle of using every bit of the data to learn about the system.

---

[1] 1% of the yearly energy cost of the US manufacturing sector is roughly a billion dollar [link, link].

Robust and computationally efficient probabilistic generative models are the crux of many machine learning applications. They are especially one of the important bottlenecks in MBRL (Deisenroth & Rasmussen, 2011; Ke et al., 2019; Chatzilygeroudis et al., 2020). System modelling for MBRL is essentially a supervised learning problem with AutoML (Zhang et al., 2021): models need to be retrained and, if needed, even retuned hundreds of times, on different distributions and data sets whose size may vary by orders of magnitude, with little human supervision. That said, there is little prior work on **rigorous comparison of system modelling algorithms**. Models are often part of a larger system, experiments are slow, and it is hard to know if the limitation or success comes from the model or from the control learning algorithm. System modelling is hard because i) data sets are non-i.i.d., and ii) classical metrics on static data sets may not be predictive of the performance on the dynamic system. There is no canonical data-generating distribution as assumed in the first page of machine learning textbooks, which makes it hard to adopt the classical train/test paradigm. At the same time, predictive system modelling is a great playground and it can be considered as an instantiation of **self-supervised learning** which some consider the "greatest challenge in ML and AI of the next few years".[2]

We propose to **compare popular probabilistic models on the Acrobot system** to study the model properties required to achieve state-of-the-art performances. We believe that such ablation studies are missing from existing "horizontal" benchmarks where the main focus is on state-of-the-art combinations of models and planning strategies (Wang et al., 2019). We start from a family of flexible probabilistic models, **autoregressive mixtures learned by deep neural nets (DARMDN)** (Bishop, 1994; Uria et al., 2013) and assess the performance of its models when removing autoregressivity, multimodality, and heteroscedasticity. We favor this family of models as it is easy i) to compare them on static data since they come with exact likelihood, ii) to simulate from them, and iii) to incorporate prior knowledge on feature types. Their greatest advantage is modelling flexibility: they can be trained with a loss allowing heteroscedasticity and, unlike Gaussian processes (Deisenroth & Rasmussen, 2011; Deisenroth et al., 2014), deterministic neural nets (Nagabandi et al., 2018; Lee et al., 2019), multivariate Gaussian mixtures (Chua et al., 2018), variational autoencoders (VAE) (Kingma & Welling, 2014; Rezende et al., 2014), and normalizing flows (Rezende & Mohamed, 2015), deep (autoregressive) mixture density nets can naturally and **effortlessly represent a multimodal posterior predictive** and what we will call $y$-*interdependence* (dependence among system observables even after conditioning on the history).

We chose Acrobot with continuous rewards (Sutton, 1996; Wang et al., 2019) which we could call the "MNIST of MBRL" for three reasons. First, it is simple enough to answer experimental questions rigorously yet it exhibits some properties of more complex environments so we believe that our findings will contribute to solve higher dimensional systems with better sample efficiency as well as better understand the existing state-of-the-art solutions. Second, Acrobot is one of the systems where i) random shooting applied on the real dynamics is state of the art in an experimental sense and ii) random shooting combined with good models is the best approach among MBRL (and even model-free) techniques (Wang et al., 2019). This means that by matching the optimal performance, **we essentially "solve" Acrobot with a sample complexity which will be hard to beat**. Third, using a single system allows both a deeper and simpler investigation of what might explain the success of popular methods. Although studying scientific hypotheses on a single system is not without precedence (Abbas et al., 2020), we leave open the possibility that our findings are valid only on Acrobot (in which case we definitely need to understand what makes Acrobot special).

There are three complementary explanations why model limitations lead to suboptimal performance in MBRL (compared to model-free RL). First, MBRL learns fast, but it converges to suboptimal models because of the lack of exploration down the line (Schaul et al., 2019; Abbas et al., 2020). We argue that there might be a second reason: the lack of the approximation capacity of these models. The two reasons may be intertwined: not only do we require from the model family to contain the real system dynamics, but we also want it to be able to represent posterior predictive distributions, which i) are consistent with the limited data used to train the model, ii) are consistent with (learnable) physical constraints of the system, and iii) allow efficient exploration. This is not the "classical" notion of approximation, it may not be alleviated by simply adding more capacity to the function representation; it needs to be tackled by properly defining the *output* of the model. Third, models are trained to predict the system one step ahead, while the planners need unbiased multi-step predictions

---

[2]https://www.facebook.com/722677142/posts/10155934004262143/

which often do not follow from one-step optimality. Our two most important findings nicely comment on these explanations.

- **Probabilistic models are needed when the system benefits from multimodal predictive uncertainty.** Although the real dynamics might be deterministic, **multimodality seems to be crucial to properly handle uncertainty around discrete jumps** in the system state that lead to qualitatively different futures.

- When systems do not exhibit such discontinuities, we do not need probabilistic predictions at all: **deterministic models are on par, in fact they consistently (although non-significantly) outperform their probabilistic versions**. We also found that heteroscedasticity at training time, perhaps acting as a regularizer, improves predictions at longer horizons (compared to classical regressors trained to minimize the mean squared error one step ahead).

Note that while our hypotheses and experimental findings are related to the grand debate on how to represent and categorize uncertainties (Deisenroth & Rasmussen, 2011; Gal, 2016; Gal et al., 2016; Depeweg et al., 2018; Osband et al., 2018; Hullermeier & Waegeman, 2019; Curi et al., 2020), we remain agnostic about which is the right representation by concentrating on *posterior* predictives on which the different approaches (e.g., Bayesian or not) are directly empirically comparable. We contribute to the debate by providing empirical evidence on a noiseless system, demonstrating unexplained phenomena even when uncertainties are purely epistemic.

We also **contribute to good practices in micro-data MBRL by building an extendable experimental protocol** in which we design static data sets and measure various metrics which may correlate with the performance of the model on the dynamic system. We instantiate the protocol by a simple setup and study models systematically in a fast experimental loop. When comparing models, the control agent or learning algorithm is part of the scoring mechanism. We fix it to a random shooting model predictive control agent, used successfully by (Nagabandi et al., 2018), for fair comparison and validation of the models. Our reproducible and extensible benchmark is made publicly available at `https://github.com/ramp-kits/rl_simulator`.

## 2 THE FORMAL SETUP

Let $\mathcal{T}_T = \big((\boldsymbol{y}_1, \boldsymbol{a}_1), \ldots, (\boldsymbol{y}_T, \boldsymbol{a}_T)\big)$ be a system trace consisting of $T$ steps of observable-action pairs $(\boldsymbol{y}_t, \boldsymbol{a}_t)$: given an observable $\boldsymbol{y}_t$ of the system state at time $t$, an action $\boldsymbol{a}_t$ was taken, leading to a new system state observed as $\boldsymbol{y}_{t+1}$. The observable vector $\boldsymbol{y}_t = (y_t^1, \ldots, y_t^{d_y})$ contains $d_y$ numerical or categorical variables, measured on the system at time $t$. The action vector $\boldsymbol{a}_t$ contains $d_a$ numerical or categorical action variables, typically set by a control function $\boldsymbol{a}_t = \pi(\mathcal{T}_{t-1}, \boldsymbol{y}_t)$ of the history $\mathcal{T}_{t-1}$ and the current observable $\boldsymbol{y}_t$ (or by a stochastic policy $\boldsymbol{a}_t \sim \pi(\mathcal{T}_{t-1}, \boldsymbol{y}_t)$).

The objective of system modelling is to predict $\boldsymbol{y}_{t+1}$ given the system trace $\mathcal{T}_t$. There are applications where point predictions $\hat{\boldsymbol{y}}_{t+1} = f(\mathcal{T}_t)$ are sufficient, however, in most control applications (e.g., reinforcement learning or Bayesian optimization) we need to access the full posterior distribution of $\boldsymbol{y}_{t+1} | \mathcal{T}_t$ to take into consideration the uncertainty of the prediction and/or to model the randomness of the system (Deisenroth & Rasmussen, 2011; Chua et al., 2018). Thus, our goal is to learn $p(\boldsymbol{y}_{t+1} | \mathcal{T}_t)$.

To convert the variable length input (condition) $\mathcal{T}_t = \big((\boldsymbol{y}_1, \boldsymbol{a}_1), \ldots, (\boldsymbol{y}_t, \boldsymbol{a}_t)\big)$ into a fixed length state vector $\boldsymbol{s}_t$ we use a fixed feature extractor $\boldsymbol{s}_t = f_{\text{FE}}(\mathcal{T}_t)$. After this step, the modelling simplifies to classical **learning of a (conditional) multi-variate density** $p(\boldsymbol{y}_{t+1} | \boldsymbol{s}_t)$ (albeit on non-i.i.d. data). In the description of our autoregressive models we will use the notation $\boldsymbol{x}_t^1 = \boldsymbol{s}_t$ and $\boldsymbol{x}_t^j = \big(y_{t+1}^1, \ldots, y_{t+1}^{j-1}, \boldsymbol{s}_t\big)$ for $j > 1$ for the input (condition) of the $j$**th autoregressive predictor** $p_j(y_{t+1}^j | \boldsymbol{x}_t^j)$. See Appendix A for more details on the autoregressive setup.

### 2.1 MODEL REQUIREMENTS

We define seven properties of the model $p$ that are desirable if to be used in MBRL. These restrict and rank the family of density estimation algorithms to consider. Req (R1) is absolutely mandatory for trajectory-sampling controllers, and Req (R2) is mandatory in this paper for using our experimental toolkit to its full extent. Reqs (R3) to (R7) are softer requirements which i) qualitatively indicate the potential performance of generative models in dynamic control, and/or ii) favor practical usability on real engineering systems and benchmarks. Table 1 provides a summary on how the different models

satisfy (or not) these requirements. We note that depending on the application and the desired control frequency of the system, one may also require models with fast prediction times.

(R1) It should be **computationally easy to properly simulate observables** $\boldsymbol{Y}_{t+1} \sim p(\cdot|\mathcal{T}_t)$ given the system trace to interface with popular control techniques that require such simulations. Note that it is then easy to obtain random traces of arbitrary length from the model by applying $p$ and $\pi$ alternately.

(R2) Given $\boldsymbol{y}_{t+1}$ and $\mathcal{T}_t$, it should be **computationally easy to evaluate** $p(\boldsymbol{y}_{t+1}|\mathcal{T}_t)$ **to obtain a likelihood score** in order to compare models on various traces. This means that $p(\boldsymbol{y}|\mathcal{T}_t) > 0$ and $\int p(\boldsymbol{y}|\mathcal{T}_t)\mathrm{d}\boldsymbol{y} = 1$ should be assured by the representation of $p$, without having to go through sampling, approximation, or numerical integration.

(R3) We should be able to **model $y$-interdependence: dependence among the $d_{\mathbf{y}}$ elements of** $\boldsymbol{y}_{t+1} = (y_{t+1}^1, \ldots, y_{t+1}^{d_y})$ **given** $\mathcal{T}_t$. In our experiments we found that the MBRL performance was not affected by the lack of this property, however, we favor it since the violation of strong physical constraints in telecommunication or robotics may hinder the acceptance of the models (simulators) by system engineers. See Appendix B for further explanation.

(R4) **Heteroscedastic** models are able to vary their uncertainty estimate as a function of the state or trace $\mathcal{T}_t$. Abbas et al. (2020) show how to use input-dependent variance to improve the planning. We found that even when using the deterministic prediction at planning time, allowing **heteroscedasticity at *training* time alleviates error accumulation down the horizon**.

(R5) Allowing **multi-modal posterior predictives** seems to be crucial to properly handle uncertainty around discrete jumps in the system state that lead to qualitatively different futures.

(R6) We should be able to **model different observable types**, for example discrete/continuous, finite/infinite support, positive, heavy tail, multimodal, etc. Engineers often have strong prior knowledge on distributions that should be used in the modelling, and the popular (multivariate) Gaussian assumption often leads to suboptimal approximation.

(R7) Complex multivariate density estimators rarely work out of the box on a new system. We are aiming at **reusability** of our models (not simple reproducibility of our experimental results). In the system modelling context, density estimators need to be retrained and retuned automatically. Both of these require **robustness and debuggability**: self-tuning and gray-box models and tools that can help the modeler to pinpoint where and why the model fails. This requirement is similar to what is often imposed on supervised models by application constraints, for example, in health care (Caruana et al., 2015).

## 2.2 EVALUATION METRICS

We define a set of metrics to compare system models both on fixed static traces $\mathcal{T}$ (Section 2.2.1) and on dynamic systems (Section 2.2.2). We have a triple aim. First, we contribute to moving the RL community towards a supervised-learning-like **rigorous evaluation** process where claims can be made more precise. Second, we define an experimental process where **models can be evaluated rapidly using static metrics** before having to run long experiments on the dynamic systems. Our methodological goal is to identify static metrics that predict the performance of the models on the dynamic system. Third, we provide diagnostics tools to the practical modeller to debug the models and define triggers and alarms when something goes wrong on the dynamical system (e.g., individual outliers, low probability traces).

### 2.2.1 STATIC METRICS

We use four metrics on our static "supervised" experiment to assess the models $p(\boldsymbol{y}_{t+1}|\boldsymbol{s}_t)$. We define all metrics formally in Appendix C. First we compute the (average) log-likelihood of $p$ on a test trace $\mathcal{T}_T$ for those models that satisfy Req (R2). Log-likelihood is a unitless metrics which is hard to interpret and depends on the unit in which its input is measured. To have a better interpretation, we normalize the likelihood with a baseline likelihood of a multivariate independent unconditional Gaussian, to obtain the **likelihood ratio (LR)** metrics. **LR is between 0 (although LR $< 1$ usually indicates a bug) and $\infty$, the higher the better.** We found that LR works well in an i.i.d. setup but distribution shift often causes "misses": test points with extremely low likelihood. Since these points dominate LR, we decided to clamp the likelihood and compute the rate of test points with a likelihood

less than[3] $p_{\min} = 1.47 \times 10^{-6}$. This **outlier rate (OR) measures the "surprise"** of a model on trace $\mathcal{T}$. **OR is between 0 and 1, the lower the better.** Third, we compute the **explained variance (R2) to quantify the precision of the predictors**. We prefer using this metrics over the MSE because it is normalized so it can be aggregated over the dimensions of $\boldsymbol{y}$. **R2 is between 0 and 1, the higher the better.** Fourth, for models that provide marginal CDFs, we compute the **Kolmogorov-Smirnov (KS)** statistics between the uniform distribution and the quantiles of the test ground truth (under the model CDFs). Well-calibrated models have been shown to improve the performance of MBRL algorithms (Malik et al., 2019). **KS is between 0 and 1, the lower the better.**

All our density estimators are trained to predict the system one step ahead yet arguably what matters is their **performance at a longer horizon** $L$ specified by the control agent. Our models do not provide explicit likelihoods $L$ steps ahead, but we can simulate from them (following ground truth actions) and evaluate the metrics by a Monte-Carlo estimate, obtaining **long horizon metrics KS($L$) and R2($L$)**. In all our experiments we use $L = 10$ with 100 Monte Carlo traces, and, for computational reasons, sample the test set at 100 random positions, which explains the high variance on these scores.

### 2.2.2 DYNAMIC METRICS

Our ultimate goal is to develop good models for MBRL so we also measure model quality in terms of the final performance. For this, **we fix the control algorithm to random shooting (RS)** (Richards, 2005; Rao, 2010) which performs well on the true dynamics of Acrobot as well as many other systems (Wang et al., 2019). RS consists in a random search of the action sequence maximizing the expected cumulative reward over a fixed planning horizon $L$. The agent then applies the first action of the best action sequence. We use $L = 10$ and generate $n = 100$ random action sequences for the random search. For stochastic models we average the cumulative rewards of 5 random trajectories obtained for a same action sequence. We note that one could achieve better results by using a larger $n$ or the cross entropy method (CEM) (de Boer et al., 2004; Chua et al., 2018). One could also consider more complex planning strategies (Wang & Ba, 2020; Argenson & Dulac-Arnold, 2020). However we judge RS with $n = 100$ to be sufficient for our study (see Appendix D for more details). We present here the MBRL loop and notations which will be needed to define the dynamic metrics.

1. Run random policy $\pi^{(1)}$ for $T = 200$ steps, starting from an initial "seed" trace $\mathcal{T}_{T_0}^{(0)}$ (typically a single-step state $\mathcal{T}_1^{(0)} = (\boldsymbol{y}_0, \cdot)$) to obtain a random initial trace $\mathcal{T}_T^{(1)}$. Let the epoch index be $\tau = 1$.

2. Learn $p^{(\tau)}$ on the full trace $\mathcal{T}_{\tau \times T} = \cup_{\tau'=1}^{\tau} \mathcal{T}_T^{(\tau')}$.

3. Run RS policy $\pi^{(\tau)}$ using model $p^{(\tau)}$, (re)starting from $\mathcal{T}_{T_0}^{(0)}$, to obtain trace $\mathcal{T}_T^{(\tau+1)}$.

4. If $\tau < N$, let $\tau = \tau + 1$ and go to Step 2, otherwise stop.

Given the formal algorithm, we can now elaborate what we mean by **system modelling for MBRL being essentially a supervised learning problem with AutoML** (and why (R7) is important). Zhang et al. (2021) make a similar argument in paper that came out independently of ours. In Step 2, the chosen model needs to be retrained and, if needed, retuned, on data sets $\mathcal{T}_{\tau \times T}$ of different distribution whose size may vary by orders of magnitude, with little human supervision. This does not mean we need to do full hyperopt in every episode $\tau$, rather that $p^{(\tau)}$ should be robust: trainable without human babysitting over a range of different distributions and data sizes. A single catastrophic learning failure (e.g. getting stuck in initial random function) means the full MBRL loop goes off the rail. Models that need to be retuned (because of sensitivity to hyperparameters) must have the retuning (AutoML) feature encapsulated into their training. The models that ended up on the top were not sensitive to the choice of hyperparameters, so we did not need to retune them in every iteration.

**MEAN ASYMPTOTIC REWARD (MAR) AND RELATIVE MAR (RMAR).** Given a trace $\mathcal{T}_T$ and a reward $r_t$ obtained at each step $t$, we define the mean reward as $R(\mathcal{T}_T) = \frac{1}{T} \sum_{t=1}^{T} r_t$.[4] The mean reward in iteration $\tau$ is then $MR(\tau) = R\left(\mathcal{T}_T^{(\tau)}\right)$. Our measure of asymptotic performance, the **mean asymptotic reward**, is the mean reward in the second half of the epochs (after convergence; we set $N$

---

[3]As a salute to 5-sigma, using the analogy of the MBRL loop (Section 2.2.2) as the iterated scientific method.

[4]The **common practice is not to normalize the cumulative reward by the (maximum) episode length** $T$, which makes it difficult to immediately compare results across papers and experiments. In micro-data RL, where $T$ is a hyperparameter (vs. part of the experimental setup), we think this should be the common practice.

in such a way that the algorithms converge after less than $N/2$ epochs) $\text{MAR} = \frac{2}{N} \sum_{\tau=N/2}^{N} \text{MR}(\tau)$. To normalize across systems and to make the measure independent of the control algorithm we use on top of the model, we define the **relative mean asymptotic reward** $\text{RMAR} = (\text{MAR} - \text{MAR}_{\text{ran}})/(\text{MAR}_{\text{opt}} - \text{MAR}_{\text{ran}})$, where $\text{MAR}_{\text{opt}}$ is the mean asymptotic reward obtained by running the same control algorithm on the true dynamics ($\text{MAR}_{\text{opt}} = 2.104$ in our experiments on Acrobot[5]), and $\text{MAR}_{\text{ran}}$ is the mean asymptotic reward obtained by running the initial random policy on the true dynamics ($\text{MAR}_{\text{ran}} = 0.12$ in our experiments on Acrobot). This puts **RMAR between 0 and 1 (the higher the better)**.

**MEAN REWARD CONVERGENCE PACE (MRCP(70)).** To assess the speed of convergence, we define the **mean reward convergence pace MRCP($p\%$)** as the number of steps needed to achieve $p\%$ of $(\text{MAR}_{\text{opt}} - \text{MAR}_{\text{ran}})$ using the running average of 5 epochs $\text{MRCP}(p\%) = T \times \arg\min_{\tau} \left( \frac{1}{5} \sum_{\tau'=\tau-2}^{\tau+2} \text{MR}(\tau) - \text{MAR}_{\text{ran}} > p\% \times (\text{MAR}_{\text{opt}} - \text{MAR}_{\text{ran}}) \right)$. The **unit of MRCP($p\%$) is system access steps**, not epochs, first to make it invariant to epoch length, and second because in micro-data RL the unit of cost is a system access step. We use $p = 70$ in our experiments.

## 2.3 THE EVALUATION ENVIRONMENT

The Acrobot benchmark system has four observables $\boldsymbol{y} = [\theta_1, \theta_2, \dot{\theta}_1, \dot{\theta}_2]$; $\theta_1$ the angle to the vertical axis of the upper link; $\theta_2$ the angle of the lower link relative to the upper link, both being normalized to $[-\pi, \pi]$; $\dot{\theta}_1$ and $\dot{\theta}_2$ the corresponding angular momenta. The action is a discrete torque on the lower link $a \in \{-1, 0, 1\}$. We use only $\boldsymbol{y}_t$ as the input to the models but augment it with the sines and cosines of the angles, so $\boldsymbol{s}_t = [\theta_1, \sin\theta_1, \cos\theta_1, \theta_2, \sin\theta_2, \cos\theta_2, \dot{\theta}_1, \dot{\theta}_2]_t$. The reward is the height of the tip of the lower link over the hanging position $r(\boldsymbol{y}) = 2 - \cos\theta_1 - \cos(\theta_1 + \theta_2) \in [0, 4]$.

We use two versions of the system to test various properties of the system models we describe in Section 3. In the "**raw angles**" system we keep $\boldsymbol{y}$ as the prediction target which means that models have to deal with the noncontinuous angle trajectories when the links roll over at $\pm\pi$. This requires multimodal posterior predictives illustrated in Figure 1 and in Appendix F. In the "**sincos**" system we change the target to $\boldsymbol{y} = [\sin\theta_1, \cos\theta_1, \sin\theta_2, \cos\theta_2, \dot{\theta}_1, \dot{\theta}_2]$ which are the observables of the Acrobot system implementation in OpenAI Gym (Brockman et al., 2016). This smoothes the target but introduces a strong nonlinear dependence between $\sin\theta_{t+1}$ and $\cos\theta_{t+1}$, even given the state $\boldsymbol{s}_t$.

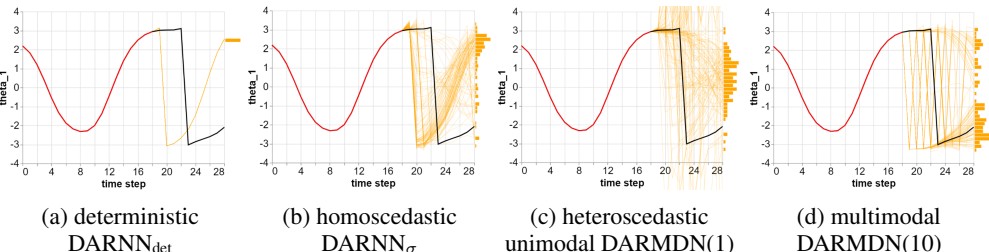

| (a) deterministic DARNN$_{\text{det}}$ | (b) homoscedastic DARNN$_\sigma$ | (c) heteroscedastic unimodal DARMDN(1) | (d) multimodal DARMDN(10) |

Figure 1: How different model types deal with uncertainty and chaos around the non-continuity at $\pm\pi$ on the Acrobot "raw angles" system. The acrobot is standing up at step 18 and hesitates whether to stay left ($\theta_1 > 0$) or go right ($\theta_1 < 0$ with a jump of $2\pi$). Deterministic and homoscedastic models underestimate the uncertainty so a small one-step error leads to picking the wrong mode and huge errors down the horizon. A heteroscedastic unimodal model correctly determines the large uncertainty but represents it as a single Gaussian so futures are not sampled from the modes. The multimodal model correctly represents the uncertainty (two modes, each with small sigma) and leads to a reasonable posterior predictive after ten steps. The thick curve is the ground truth, the red segment is past, the black segment is future, and the orange curves are simulated futures. See Section 3 for the definition of the different models and Appendix F for more insight.

Our aim of predicting dynamic performance on static experiments will require not only score design but also **data set design.** In this paper we evaluate our models on **two data sets**. The first is generated by **running a random policy** $\pi^{(1)}$ on Acrobot. We found that this was too easy to learn, so scores hardly predicted the dynamic performance of the models (Schaul et al., 2019). To create a more

---

[5]See Table 5 in Appendix D for more discussion on $\text{MAR}_{\text{opt}}$.

"skewed" data set, we execute the MBRL loop (Section 2.2.2) for **one iteration using the linear ARLin**$_\sigma$ model (see Section 3), and generate traces using the resulting policy $\pi^{(2)}_{\text{ARLin}_\sigma}$. On both data sets we use ten-fold cross validation on 5K training points and report test scores on a held-out test set of 20K points. All sets comprise of episodes of length 500, starting from an approximately hanging position: all state variables (the angles and the angular velocities) are uniformly sampled in $[-0.1, 0.1]$.

## 3 MODELS AND RESULTS

A commonly held belief (Lee et al., 2019; Wang et al., 2019) is that MBRL learns fast but cannot reach the asymptotic performance of model-free RL. It presumes that models either "saturate" (their approximation error cannot be eliminated even when the size of the training set grows high) and/or they get stuck in local minima (since sampling and learning are coupled). Our research goal is to design models that alleviate these limitations. The first step is to introduce and study models that are learnable with small data but are flexible enough to represent complicated functions (see the summary in Table 1). Implementation details are given in Appendix D.

Table 1: Summary of the different models satisfying (or not) the various requirements from Section 2.1. (R1): efficient simulation; (R2): explicit likelihood; (R3): $\boldsymbol{y}$-interdependence (yellow means "partially"); (R4): heteroscedasticity (yellow means "at training"); (R5): multimodality (yellow means "in principle, yes, in practice, no"); (R6): ability to model different feature types; (R7): robustness and debuggability. The last two columns indicate whether the model is among the optimal ones on the Acrobot sincos and raw angles systems (Section 2.3 and Table 2; yellow means significantly worse than the best model but within 5% of the optimum).

| Model | (R1) | (R2) | (R3) | (R4) | (R5) | (R6) | (R7) | sincos | raw angles |
|---|---|---|---|---|---|---|---|---|---|
| ARLin$_\sigma$ | ✓ | ✓ | ✓ | | | ✓ | ✓ | | |
| DARNN$_\sigma$ | ✓ | ✓ | ✓ | | | ✓ | ✓ | ✓ | |
| GP | ✓ | ✓ | | ✓ | | | | ✓ | |
| DMDN(1) | ✓ | ✓ | | ✓ | | ✓ | | ✓ | |
| DMDN(10) | ✓ | ✓ | ✓ | ✓ | ✓ | ✓ | | ✓ | ✓ |
| DARMDN(1) | ✓ | ✓ | ✓ | ✓ | | ✓ | ✓ | ✓ | |
| DARMDN(10) | ✓ | ✓ | ✓ | ✓ | ✓ | ✓ | ✓ | ✓ | ✓ |
| PETS (bagged DMDN(1)) | ✓ | ✓ | | ✓ | ✓ | ✓ | | ✓ | |
| VAE | ✓ | | | ✓ | ✓ | ✓ | | ✓ | |
| RealNVP | ✓ | ✓ | | ✓ | ✓ | | | | |
| DARNN$_{\text{det}}$ | ✓ | | ✓ | | | ✓ | ✓ | ✓ | |
| DMDN(1)$_{\text{det}}$ | ✓ | | | ✓ | | ✓ | ✓ | ✓ | |
| DARMDN(1)$_{\text{det}}$ | ✓ | | ✓ | ✓ | | ✓ | ✓ | ✓ | |

**AUTOREGRESSIVE DETERMINISTIC REGRESSOR + FIXED VARIANCE.** We learn $d_{\text{y}}$ **deterministic regressors** $f_1(\boldsymbol{x}^1), \ldots, f_{d_{\text{y}}}(\boldsymbol{x}^{d_{\text{y}}})$ by minimizing MSE and estimate a **uniform residual variance** $\sigma_j^2 = \frac{1}{T-2} \sum_{t=1}^{T-1} \left( y_{t+1}^j - f_j(\boldsymbol{x}_t^j) \right)^2$ for each output dimension $j = 1, \ldots, d_{\text{y}}$. The probabilistic model is then Gaussian $p_j(y^j | \boldsymbol{x}^j) = \mathcal{N}\left( y^j; f_j(\boldsymbol{x}^j), \sigma_j \right)$. The two baseline models of this type are **linear regression (ARLin**$_\sigma$**)** and a **neural net (DARNN**$_\sigma$**)**. These models are easy to train, they can handle $\boldsymbol{y}$-interdependence (since they are autoregressive), but they fail (R5) and (R4): they cannot handle multimodal posterior predictives and heteroscedasticity.

**GAUSSIAN PROCESS (GP)** is the method of choice in the popular PILCO algorithm (Deisenroth & Rasmussen, 2011). On the modelling side, it cannot handle non-Gaussian (multimodal or heteroscedastic) posteriors and $\boldsymbol{y}$-interdependence, failing Req (R6). More importantly, similarly to Wang et al. (2019) and Chatzilygeroudis et al. (2020), we found it very hard to tune and slow to simulate from. We have reasonable performance on the sincos data set which we report, however GPs failed the raw angles data set (as expected due to angle non-continuity) and, more importantly, the hyperparameters tuned lead to suboptimal dynamical performance, so we decided not to report these results. We believe that generative neural nets that can learn the same model family are more robust, faster to train and sample from, and need less babysitting in the MBRL loop.

**MIXTURE DENSITY NETS.** A classical **deep mixture density net DMDN($D$)** (Bishop, 1994) is a feed-forward neural net outputting $D(1 + 2d_y)$ parameters $[w^\ell, \boldsymbol{\mu}^\ell, \boldsymbol{\sigma}^\ell]_{\ell=1}^D$, $\boldsymbol{\mu}^\ell = [\mu_j^\ell]_{j=1}^{d_y}$, $\boldsymbol{\sigma}^\ell = [\sigma_j^\ell]_{j=1}^{d_y}$ of a multivariate independent Gaussian mixture $p(\boldsymbol{y}|\boldsymbol{s}) = \sum_{\ell=1}^D w^\ell(\boldsymbol{s})\mathcal{N}(\boldsymbol{y}; \boldsymbol{\mu}^\ell(\boldsymbol{s}), \mathrm{diag}(\boldsymbol{\sigma}^\ell(\boldsymbol{s})^2))$. Its **autoregressive counterpart DARMDN($D$)** learns $d_y$ independent neural nets outputting the $3Dd_y$ parameters $[w_j^\ell, \mu_j^\ell, \sigma_j^\ell]_{j,\ell}$ of $d_y$ mixtures $p_1, \ldots, p_{d_y}$ (2). Both models are trained to maximize the log likelihood (3). They can both represent heteroscedasticity and, for $D > 1$, multimodal posterior predictives. In engineering systems we prefer DARMDN for its better handling of $\boldsymbol{y}$-interdependence and its ability to model different types of system variables. DARMDN($D$) is similar to RNADE (Uria et al., 2013) except that in system modelling we do not need to couple the $d_y$ neural nets. While RNADE was used for anomaly detection (Iwata & Yamanaka, 2019), acoustic modelling (Uria et al., 2015), and speech synthesis (Wang et al., 2017), to our knowledge, neither DARMDN nor RNADE have been used in the context of MBRL. DMDN has been used in robotics by Khansari-Zadeh & Billard (2011) and it is an

Table 2: Model evaluation results on the dynamic environments using random shooting MPC agents. RMAR is the percentage of the optimum reward achieved asymptotically, and MRCP(70) is the number of system access steps needed to achieve 70% of the optimum reward (Section 2.2.2). ↓ and ↑ mean lower and higher the better, respectively. Unit is given after the / sign.

| Method | RMAR/$10^{-3}$↑ | MRCP(70)↓ |
|---|---|---|
| | Acrobot raw angles system | |
| ARLin$_\sigma$ | 215±7 | NaN±NaN |
| DARNN$_\sigma$ | 612±9 | 14070±3350 |
| DARNN$_{det}$ | 703±7 | 5660±980 |
| **DMDN(10)** | **968±8** | 2200±240 |
| DARMDN(1) | 730±7 | 3320±680 |
| **DARMDN(10)** | **963±7** | **1680±100** |
| DARMDN(10)$_{det}$ | 709±7 | 3960±570 |
| PETS | 715±7 | 7260±2200 |
| VAE | 668±11 | 15100±3450 |
| | Acrobot sincos system | |
| ARLin$_\sigma$ | -11±3 | NaN±NaN |
| DARNN$_\sigma$ | 947±8 | 1600±170 |
| DARNN$_{det}$ | 963±8 | 1440±80 |
| **DMDN(10)** | **980±8** | 1670±90 |
| **DARMDN(1)** | **982±7** | 1400±50 |
| DARMDN(10) | 977±8 | 1340±100 |
| **DARMDN(10)$_{det}$** | **986±7** | 1300±100 |
| **DARMDN(1)$_{det}$** | **987±7** | 1300±80 |
| **PETS** | **992±7** | 1040±110 |
| **PETS$_{det}$** | **995±7** | **840±40** |
| VAE | 952±10 | 1770±190 |
| RealNVP | 536±27 | NaN±NaN |

important brick in the world model of Ha & Schmidhuber (2018). **Probabilistic Ensembles with Trajectory Sampling (PETS)** (Chua et al., 2018) is an important contribution to MBRL that trains a DMDN($D$) model by bagging $D$ DMDN(1) models. In our experiments we also found that bagging can improve the LR score (4) significantly, and bagging seems to accelerate learning by being more robust for small data sets (MRCP(70) score in Table 2 and learning curves in Appendix E); however bagged single Gaussians are not multimodal (all bootstrap samples will pick instances from every mode) so PETS fails on the raw angles data.

**DETERMINISTIC MODELS** are important baselines, used successfully by Nagabandi et al. (2018) and Lee et al. (2019) in MBRL. They fail Req (R2) but can be alternatively validated using R2. On the other hand, when **used in an autoregressive setup**, if the mean prediction represents the posterior predictives well (unimodal distributions with small uncertainty), they work well. In fact, in our experiments we found that **deterministic models are consistently (although non-significantly) better than their probabilistic versions**, possibly because the mean prediction is more precise. We implemented deterministic models by "sampling" the mean of the DARNN$_\sigma$ and DARMDN($\cdot$) models, obtaining **DARNN$_{det}$** and **DARMDN($\cdot$)$_{det}$**, respectively.

**VARIATIONAL AUTOENCODERS AND FLOWS.** We tested two other popular techniques, variational autoencoders (VAE) (Kingma & Welling, 2014; Rezende et al., 2014) and the flow-based RealNVP (Dinh et al., 2017). VAE does not provide exact likelihood (R2); RealNVP does, but the R2 and KS scores are harder to compute. In principle they can represent multimodal posterior predictives, but **in practice they do not seem to be flexible enough to work well on the raw angles system**. A potential solution would be to enforce a multimodal output as done by Moerland et al. (2017). VAE performed well (although significantly worse than the mixture models) on the sincos system.

Our results are summarized in Tables 2 and 3. We show mean reward learning curves in Appendix E. We found that comparing models solely based on their performance on the random policy data is a bad choice: most models did well in both the raw angles and sincos systems. Static **performance on the linear policy data is a better predictor** of the dynamic performance; among the scores, not surprisingly, and also noted by Nagabandi et al. (2018), the **R2(10) score correlates the most with dynamic performance**.

Table 3: Model evaluation results on static data sets. ↓ and ↑ mean lower and higher the better, respectively. Unit is given after the / sign.

| Method | LR↑ | OR/$10^{-4}$↓ | R2/$10^{-4}$↑ | KS/$10^{-3}$↓ | R2(10)/$10^{-4}$↑ | KS(10)/$10^{-3}$↓ | trt/min↓ | tst/sec↓ |
|---|---|---|---|---|---|---|---|---|
| | | | | Acrobot raw angles, data generated by random policy | | | | |
| ARLin$_\sigma$ | 27±1 | 44±7 | 9763±0 | 177±3 | 8308±485 | 157±11 | 0±0 | 0±0 |
| DARNN$_\sigma$ | 54±8 | 171±37 | 9829±9 | 171±36 | 8711±491 | 212±48 | 2±0 | 1±0 |
| DMDN(10) | 430±26 | 0±0 | 9790±2 | 124±10 | 8973±456 | 129±29 | 15±0 | 2±0 |
| DARMDN(1) | 424±18 | 10±2 | 9784±2 | 126±6 | 9267±269 | 106±17 | 19±0 | 2±0 |
| DARMDN(10) | 410±8 | 3±1 | 9782±2 | 135±8 | 9049±375 | 122±17 | 18±0 | 2±0 |
| | | | | Acrobot raw angles, data generated by linear policy after one epoch | | | | |
| ARLin$_\sigma$ | 3±0 | 20±5 | 6832±9 | 85±1 | 398±270 | 87±14 | 0±0 | 0±0 |
| DARNN$_\sigma$ | 25±1 | 176±31 | 9574±13 | 193±16 | 4844±477 | 139±23 | 2±0 | 1±0 |
| DMDN(10) | 137±10 | 40±11 | 8449±443 | 72±9 | 5659±1086 | 135±19 | 15±0 | 2±0 |
| DARMDN(1) | 120±2 | 56±12 | 5677±6 | 47±5 | 1291±846 | 114±20 | 20±1 | 2±0 |
| DARMDN(10) | 143±6 | 22±6 | 9571±70 | 62±5 | 8065±363 | 100±11 | 20±0 | 2±0 |
| | | | | Acrobot sincos, data generated by random policy | | | | |
| ARLin$_\sigma$ | 6±0 | 47±10 | 8976±1 | 118±3 | 5273±320 | 110±11 | 0±0 | 0±0 |
| DARNN$_\sigma$ | 50±4 | 188±20 | 9987±5 | 176±22 | 9249±623 | 257±64 | 4±0 | 2±0 |
| GP | 88±2 | 0±0 | 9999±0 | 224±11 | 9750±85 | 168±29 | 0±0 | 9±1 |
| DMDN(10) | 361±22 | 0±0 | 9957±4 | 139±15 | 8963±538 | 146±35 | 21±1 | 1±0 |
| DARMDN(1) | 281±5 | 3±1 | 9950±5 | 151±3 | 8953±337 | 131±18 | 27±1 | 3±0 |
| DARMDN(10) | 288±7 | 1±0 | 9983±4 | 153±10 | 9296±233 | 140±25 | 28±1 | 4±1 |
| | | | | Acrobot sincos, data generated by linear policy after one epoch | | | | |
| ARLin$_\sigma$ | 2±0 | 11±4 | 6652±9 | 46±1 | 354±304 | 127±18 | 0±0 | 0±0 |
| DARNN$_\sigma$ | 32±2 | 166±34 | 9986±2 | 156±16 | 7944±1061 | 194±29 | 4±0 | 2±0 |
| GP | 56±1 | 6±1 | 9995±0 | 113±4 | 8334±185 | 133±15 | 0±0 | 9±1 |
| DMDN(10) | 95±5 | 29±6 | 9993±1 | 85±9 | 9001±285 | 128±17 | 21±0 | 1±0 |
| DARMDN(1) | 125±4 | 12±4 | 9991±1 | 80±4 | 8693±286 | 89±13 | 32±2 | 3±0 |
| DARMDN(10) | 119±4 | 9±5 | 9991±2 | 68±4 | 8655±269 | 95±15 | 30±1 | 4±0 |

Our most counter-intuitive result (although Wang et al. (2019) and Wang & Ba (2020) observed a similar phenomenon) is that DARMDN($\cdot$)$_{\text{det}}$ and PETS$_{\text{det}}$ are tied for winning on the sincos system, which suggests that a **deterministic model can be on par with (or even slightly better than) the best probabilistic models** if the system requires no multimodality. What is even more surprising is that classical neural net DARNN$_{\text{det}}$ is slightly but significantly worse, suggesting that **the optimal model, even if it is deterministic, needs to be trained for a likelihood score in a generative setup**. The lower R2(10) score of DARNN$_{\text{det}}$ (and the case study in Appendix F) suggest that classical regression optimizing MSE leads to error accumulation and thus subpar performance down the horizon. Our hypothesis is that heteroscedasticity at training time acts as a regularizer, leading somehow to less error accumulation at a longer horizon.

On the sincos system PETS reaches the optimum MAR$_{\text{opt}}$ within statistical uncertainty which means that this setup of the Acrobot system is essentially solved. We improve the convergence pace MCPR(70) of the PETS implementation of Wang & Ba (2020) by **two to four folds** (Figure 3 in Appendix E) by using a more ambitious learning schedule (short epochs and frequent retraining). The real forte of D(AR)MDN(10) is the 95% RMAR score on the raw angles system that requires multimodality, **beating the other models by more than 20%**. It suggests remarkable robustness that makes it the method of choice for larger systems with more complex dynamics.

## 4 CONCLUSION AND FUTURE WORK

Our study was made possible by developing a toolbox of good practices for model evaluations and debuggability in model-based reinforcement learning, particularly useful when trying to solve real world applications with domain engineers. We found that heteroscedasticity at *training time* alleviates error accumulation down the horizon. Then at *planning time*, we do not need stochastic models: the deterministic mean prediction suffices. That is, unless the system requires multimodal posterior predictives, in which case deep (autoregressive or not) mixture density nets are the only current generative models that work. Our findings lead to state-of-the-art sample complexity (by far) on the Acrobot system by applying an aggressive training schedule. The most important future direction is to extend the results to more complex systems requiring larger planning horizons and to planning strategies beyond random shooting.

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

## A   AUTOREGRESSIVE MIXTURE DENSITIES

The multi-variate density $p(\boldsymbol{y}_{t+1}|\boldsymbol{s}_t)$ is decomposed into a **chain of one-dimensional densities**

$$p(\boldsymbol{y}_{t+1}|\boldsymbol{s}_t) = p_1(y_{t+1}^1|\boldsymbol{s}_t)\prod_{j=2}^{d_y} p_j(y_{t+1}^j|y_{t+1}^1,\ldots,y_{t+1}^{j-1},\boldsymbol{s}_t) = p_1(y_{t+1}^1|\boldsymbol{x}_t^1)\prod_{j=2}^{d_y} p_j(y_{t+1}^j|\boldsymbol{x}_t^j), \quad (1)$$

where, for simplicity, **we denote the input (condition) of the $j$th autoregressive predictor by** $\boldsymbol{x}_t^j = (y_{t+1}^1,\ldots,y_{t+1}^{j-1},\boldsymbol{s}_t)$. First, $p$ is a proper $d_y$-dimensional density as long as the components $p_j$ are valid one-dimensional densities (Req (R2)). Second, if it is easy to draw from the components $p_j$, it is easy to simulate $\boldsymbol{Y}_{t+1}$ following the order of the chain (1) (Req (R1)). Third, Req (R3) is satisfied by construction. But the real advantages are on the logistics of modelling. Unlike in computer vision (pixels) or NLP (words), engineering systems often have inhomogeneous features that should be modeled differently. There exists a plethora of different one-dimensional density models which we can use in the autoregressive setup, whereas multi-dimensional extensions are rare, especially when feature types are different (Req (R6)). At the debuggability side (Req (R7)) the advantage is the availability of one-dimensional goodness of fit metrics and visualization tools which make it easy to pinpoint what goes wrong if the model is not working. On the negative side, autoregression breaks the symmetry of the output variables by introducing an artificial ordering and, depending on the family of the component densities $p_j$, the modelling quality may depend on the order.

To preserve these advantages and alleviate the order dependence we found that we needed a rich family of one-dimensional densities so we decided to use mixtures

$$p_j(y^j|\boldsymbol{x}^j) = \sum_{\ell=1}^{D} w_j^\ell(\boldsymbol{x}^j) P_j^\ell\big(y^j; \theta_j^\ell(\boldsymbol{x}^j)\big), \quad (2)$$

where component types $P_j^\ell$, component parameters $\theta_j^\ell$, and component weights $w_j^\ell$ can all depend on $j$, $\ell$, and the input $\boldsymbol{x}^j$. In general, the modeller has a large choice of easy-to-fit component types to choose from given the type of variable $y^j$ (Req (R6)); in this paper all our variables were numerical so we only use Gaussian components with free mean and variance. Contrary to the widely held belief (Papamakarios et al., 2017), in our experiments **we found no evidence that the ordering of the variables matters**, arguably because of the flexibility of the one-dimensional mixture models that can pick up non-Gaussian features such as multimodality (Req (R5)). Finally a computational advantage: given a test point $\boldsymbol{x}$, we do not need to carry around (density) functions: our representation of $p(\boldsymbol{y}|\boldsymbol{x})$ is a numerical vector concatenating $\big[w_j^\ell, P_j^\ell, \theta_j^\ell\big]_{j,\ell}$.

## B   $\boldsymbol{y}$-INTERDEPENDENCE

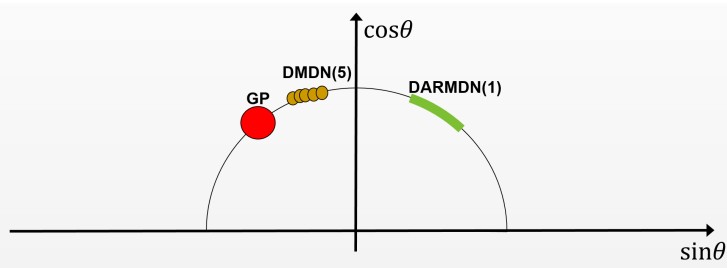

Figure 2: How different models handle $\boldsymbol{y}$-interdependence. GP (and DMDN(1)) "spreads" the uncertainty in all directions, leading to non-physical predictions. DMDN($D > 1$) may "tile" the nonlinear $\boldsymbol{y}$-interdependence with smaller Gaussians, and in the limit of $D \to \infty$ it can handle $\boldsymbol{y}$-interdependence for the price of a large number of parameters to learn. DARMDN, with its autoregressive function learning, can put the right amount of dependent uncertainty on $y^2|y^1$, learning for example the noiseless functional relationship between $\cos\theta$ and $\sin\theta$.

$y$-interdependence is the **dependence among the $d_y$ elements of $y_{t+1} = (y_{t+1}^1, \ldots, y_{t+1}^{d_y})$ given $\mathcal{T}_t$.**
Some popular algorithms such as PILCO (Deisenroth & Rasmussen, 2011) suppose that elements of
$y_{t+1}$ are independent given $\mathcal{T}_t$. It is a reasonable assumption when modelling aleatoric uncertainty in
stochastic systems with independent noise, but it is clearly wrong when the posterior predictive has a
structure due to functional dependence. It happens even in the popular AI Gym benchmark systems
(Brockman et al., 2016) (think about usual representation of angles: $\cos \theta_{t+1}$ is clearly dependent of
$\sin \theta_{t+1}$ even given $\mathcal{T}_t$; see Figure 2), let alone systems with strong physical constraints in telecom-
munication or robotics. Generating non-physical traces due to not modelling $y$-interdependence
may lead not only to subpar performance but also to reluctance to accept the models (simulators) by
system engineers.

## C STATIC METRICS

We define our static metrics from the decomposition of the multivariate density $p(y_{t+1}|s_t)$ into the
product of one-dimensional densities (see Appendix A for details):

$$p(y_{t+1}|s_t) = p_1(y_{t+1}^1|x_t^1) \prod_{j=2}^{d_y} p_j(y_{t+1}^j|x_t^j) \quad \text{where} \quad x_t^j = (y_{t+1}^1, \ldots, y_{t+1}^{j-1}, s_t).$$

**LIKELIHOOD RATIO TO A SIMPLE BASELINE (LR)** is our "master" metrics. The (average)
log-likelihood

$$\mathcal{L}(\mathcal{T}_T; p) = \frac{1}{d_y} \sum_{j=1}^{d_y} \frac{1}{T-1} \sum_{t=1}^{T-1} \log p_j \left(y_{t+1}^j | x_t^j\right) \tag{3}$$

can be evaluated easily on any trace $\mathcal{T}_T$ thanks to Req (R2). Log-likelihood is a unitless metrics
which is hard to interpret and depends on the unit in which its input is measured (this variability
is particularly problematic when $p_j$ is a mixed continuous/discrete distribution). To have a better
interpretation, we normalize the likelihood

$$\mathrm{LR}(\mathcal{T}; p) = \frac{e^{\mathcal{L}(\mathcal{T}; p)}}{e^{\mathcal{L}_b(\mathcal{T})}} \tag{4}$$

with a baseline likelihood $\mathcal{L}_b(\mathcal{T})$ which can be adapted to the feature types. In our experiments
$\mathcal{L}_b(\mathcal{T})$ is a multivariate independent unconditional Gaussian. **LR is between 0 (although LR < 1
usually indicates a bug) and $\infty$, the higher the better.**

**OUTLIER RATE (OR).** We found that LR works well in an i.i.d. setup but distribution shift often
causes "misses": test points with extremely low likelihood. Since these points dominate $\mathcal{L}$ and LR,
we decided to **clamp the likelihood at**[6] $p_{\min} = 1.47 \times 10^{-6}$. Given a trace $\mathcal{T}$ and a model $p$, we
define $\mathcal{T}(p; p_{\min}) = \{(y_t, a_t) \in \mathcal{T} : p(y_t|x_{t-1}) > p_{\min}\}$, report $\mathrm{LR}\big(\mathcal{T}(p; p_{\min}); p\big)$ instead of
$\mathrm{LR}(\mathcal{T}; p)$, and measure the "surprise" of a model on trace $\mathcal{T}$ by the **outlier rate (OR)**

$$\mathrm{OR}(\mathcal{T}; p) = 1 - \frac{|\mathcal{T}(p; p_{\min})|}{|\mathcal{T}|}. \tag{5}$$

**OR is between 0 and 1, the lower the better.**

**EXPLAINED VARIANCE (R2)** assesses the **mean performance (precision)** of the methods. For-
mally

$$\mathrm{R2}(\mathcal{T}_T; p) = \frac{1}{d_y} \sum_{j=1}^{d_y} \left(1 - \frac{\mathrm{MSE}_j(\mathcal{T}_T; p)}{\sigma_j^2}\right) \quad \text{with} \quad \mathrm{MSE}_j(\mathcal{T}_T; p) = \frac{1}{T-1} \sum_{t=1}^{T-1} \left(y_{t+1}^j - f_j(x_t)\right)^2,$$
$$\tag{6}$$

where $f_j(x_t) = \mathbb{E}_{p_j(\cdot|x_t^j)} \{y^j\}$ is the expectation of $y_{t+1}^j$ given $x_t^j$ under the model $p_j$ (point
prediction), and $\sigma_j^2$ is the sample variance of $(y_1^j, \ldots, y_T^j)$. We prefer using this metrics over the

---

[6]As a salute to five sigma, using the analogy of the MBRL loop (Section 2.2.2) being the iterated scientific
method.

MSE because it is normalized so it can be aggregated over the dimensions of $\boldsymbol{y}$. **R2 is between 0 and 1, the higher the better.**

**CALIBRATEDNESS (KS).** Well-calibrated models have been shown to improve the performance of algorithms (Malik et al., 2019). A **well-calibrated** density estimator has the property that the quantiles of the (test) ground truth are uniform. To assess this, we compute the **Kolmogorov-Smirnov (KS)** statistics. Formally, let $F_j(y^j|\boldsymbol{x}^j) = \int_{-\infty}^{y^j} p_j(y'|\boldsymbol{x}^j)\mathrm{d}y'$ be the cumulative distribution function (CDF) of $p_j$, and let the order statistics of $\mathcal{F}_j = \left[F_j\left(y_{t+1}^j|\boldsymbol{x}_t^j\right)\right]_{t=1}^{T-1}$ be $s_j$, that is, $F_j\left(y_{s_j}^j|\boldsymbol{x}_{s_j}^j\right)$ is the $s_j$th largest quantile in $\mathcal{F}_j$. Then we define

$$\mathrm{KS}(\mathcal{T}_T; F) = \frac{1}{d_\mathrm{y}} \sum_{j=1}^{d_\mathrm{y}} \max_{s_j \in [1, T-1]} \left| F_j\left(y_{s_j}^j|\boldsymbol{x}_{s_j}^j\right) - \frac{s_j}{T-1} \right|. \tag{7}$$

Computing KS requires that the model can provide conditional CDFs, which further filters the possible models we can use. On the other hand, the aggregate KS and especially the one-dimensional CDF plots ($F_j(y_{s_j}^j|\boldsymbol{x}_{s_j}^j)$ vs. $s_j/(T-1)$) are great debugging tools. **KS is between 0 and 1, the lower the better.**

All four metrics (LR, OR, R2, KS) are averaged over the dimensions, but for **debugging we can also evaluate them dimension-wise**.

**LONG HORIZON METRICS KS($L$) AND R2($L$).** All our density estimators are trained to predict the system one step ahead yet arguably **what matters is their performance at a longer horizon $L$** specified by the control agent. Our models do not provide explicit likelihoods $L$ steps ahead, but we can simulate from them (following ground truth actions) and evaluate the metrics by a Monte-Carlo estimate. Given $n$ random estimates $\mathcal{Y}_L = [\hat{\boldsymbol{y}}_{t+L,\ell}]_{\ell=1}^n$, we can use $f_j(\boldsymbol{x}_t) = \frac{1}{n}\sum_{\hat{\boldsymbol{y}} \in \mathcal{Y}_L} \hat{y}^j$ in (6) to obtain an **unbiased R2($L$) estimate**. To obtain a **KS($L$) estimate**, we order $\mathcal{Y}_L$ and approximate $F_j(y^j|\boldsymbol{x}^j)$ by $\frac{1}{n}|\{\hat{\boldsymbol{y}} \in \mathcal{Y}_L : \hat{y}^j < y^j\}|$ in (7). LR and OR would require approximate techniques so we omit them. In all our experiments we use $L = 10$, $n = 100$, and, for computational reasons, sample the test set at 100 random positions, which explains the high variance on these scores.

All six metrics (LR, OR, R2, KS, R2(10), KS(10)) are averaged over the dimensions to obtain single scores for the environment/model pair, but for **debugging we can also evaluate them dimension-wise**. LR is the "master" score that combines precision (R2) and calibratedness (KS). R2 is a good single measure to assess the models, especially when iterated to obtain R2($L$). OR and KS are excellent debugging tools. The single-target KS and quantile plots are especially useful to spot *how* the models are miscalibrated: e.g., points accumulating in the middle indicate that we overestimate the tails, leading to nonphysical simulations, and vice versa, accumulation at the edges means our model is missing modes. OR is great to detect catastrophic failures or distribution shifts, so monitoring it on the deployed system is crucial. Finally, correlating these metrics to the dynamic performance (Section 2.2.2) for the given system can form the basis of a comprehensive monitoring system which is as important as model performance in practice.

# D  IMPLEMENTATION DETAILS

Note that all experimental code is publicly available at https://github.com/ramp-kits/rl_simulator. In this section we give enough information so that all models can be reproduced by a moderately experienced machine learning expert.

The sincos and raw angles Acrobot systems are based on the OpenAI Gym implementation (Brockman et al., 2016). The starting position of each episode is the one obtained from the default `reset` function of this implementation: all state variables (the angles and the angular velocities) are uniformly sampled in $[-0.1, 0.1]$. For the linear regression model we use the implementation of Scikit-learn (Pedregosa et al., 2011) without regularization. We use Pytorch (Paszke et al., 2019) for the neural network based models (DARNN, DMDN and DARMDN) and Gpytorch (Gardner et al., 2018) for the GP models. The hyperparameter search for these models was done in two steps: first using random search over a coarse hyperparameter grid, then using a second step of random search over a finer grid around values of interest. The steps of the coarse grid were defined to contain five values of each

hyperparameters (or less where applicable), the finer grid was defined to contain five values of each hyperparameter (or less where applicable) between two interesting spots close in the hyperparameter space. The selected hyperparameters are given in Table 4.

"Nb layers" corresponds to the number of fully connected layers, except for the two following models:

- RealNVP (Dinh et al., 2017): it is the number of coupling layers.
- CVAE (Sohn et al., 2015): it is the total number of layers (encoder plus decoder).

"Nb components" is the number of components in the outputted density mixture. In the GP and deterministic NN cases, it is trivially one.

Table 4: Model hyperparameters.

| Method | Learning rate | Neurons per layer | Nb layers | Nb components | Validation size | Nb epochs |
|---|---|---|---|---|---|---|
| | | | Tried values | | | |
| DARNN$_\sigma$ | [1e-4, 1e-1] | [20, 300] | [1, 4] | 1 | [0.05, 0.4] | [10, 300] |
| DMDN | [1e-5, 1e-2] | [100, 600] | [2, 5] | [2, 20] | [0.05, 0.4] | [50, 500] |
| DARMDN | [1e-5, 1e-2] | [20, 300] | [1, 4] | [2, 20] | [0.05, 0.4] | [50, 500] |
| CVAE | [1e-5, 1e-2] | [20, 300] | [4, 10] | NaN | [0.05, 0.4] | [50, 500] |
| RealNVP | [1e-5, 1e-2] | [10, 300] | [2, 5] | NaN | [0.05, 0.4] | [50, 500] |
| GP | [1e-3, 1e-1] | NaN | NaN | 1 | [0.05, 0.4] | [10, 300] |
| | | | Best values | | | |
| DARNN$_\sigma$ | 4e-3 | 200 | 3 | 1 | 0.05 | 100 |
| DMDN(10) | 5e-3 | 200 | 3 | 10 | 0.1 | 300 |
| DARMDN(1) | 1e-3 | 50 | 3 | 1 | 0.1 | 300 |
| DARMDN(10) | 1e-3 | 100 | 3 | 10 | 0.1 | 300 |
| CVAE | 1e-3 | 60 | 4 | NaN | 0.15 | 100 |
| RealNVP | 5e-3 | 20 | 3 | NaN | 0.15 | 200 |
| GP | 5e-2 | NaN | NaN | 1 | 0.15 | 50 |

For PETS we use the code shared by Wang et al. (2019) for the Acrobot sincos system. Following Chua et al. (2018), the size of the ensemble is set to 5. For the Acrobot raw angles system we use the same PETS neural network architecture as the one available for the original sincos system. Although the default number of epochs was set to 5 in the available code we reached better results with 100 epochs and use this value in our results. Finally, the RS agent is configured to be the same as the one we use: planning horizon $L = 10$, search population size $n = 100$ and 5 particles.

We selected the planning strategy (random shooting with search population size $n = 100$) by evaluating the performance of random shooting and the cross entropy method (CEM) on the true dynamics for different values of $n$. Results are presented in Table 5. Although for both RS and CEM with $n = 500$ leads to a better performance, $n = 100$ is already sufficient to achieve more than decent mean rewards and outperform the result of Wang et al. (2019) while reducing the total computational cost of the study. CEM was implemented with a learning rate of 0.1, an elite size equal to 50 and 5 iterations. For a fair comparison between RS and CEM $n$ means the total number of sampled action sequences. This means that, for CEM, $n$ means a search population size of $n/5$ for each of the 5 iterations.

Table 5: Comparison of RS and CEM on the true dynamics. The $\pm$ values are 90% Gaussian confidence intervals based on 100 random repetitions of a 200-step rollout.

| RS with $n = 100$ | RS with $n = 500$ | RS with $n = 1000$ | CEM with $n = 500$ | CEM with $n = 1000$ |
|---|---|---|---|---|
| $2.10 \pm 0.035$ | $2.27 \pm 0.025$ | $2.29 \pm 0.024$ | $2.32 \pm 0.021$ | $2.30 \pm 0.021$ |

We implemented reusable system models and static experiments within the RAMP framework (Kégl et al., 2018).

All $\pm$ values in the results tables are 90% Gaussian confidence intervals based on i) 10-fold cross-validation for the static scores in Table 3, ii) 50 epochs and two to ten seeds in the RMAR column, and iii) ten seeds in the MRCP(70) column of Table 2.

## E    MEAN REWARD LEARNING CURVES

Figure 3 shows the mean reward learning curves on the Acrobot raw angles and sincos systems. The top models PETS and DARMDN(10)$_{det}$ converge close to the optimum at around the same pace on the sincos system. PETS converges slightly faster than the other models in the early phase. Our hypothesis is that bagging creates more robust models in the extreme low data regime (100s of training points). Our models were tuned using 5000 points which seems to coincide with the moment when the bagging advantage disappears.

On the raw angles system DARMDN(10) and DMDN(10) separate from the pack indicating that this setup requires non-deterministic predictors and mixture densities to model multimodal posterior predictives. The reward is between 0 (hanging) and 4 (standing up). Each epoch starts at hanging position and it takes about 100 steps to reach the stationary regime where the tip of acrobot is above the horizontal line most of the time. This means that reaching an average reward above 2 needs an excellent control policy.

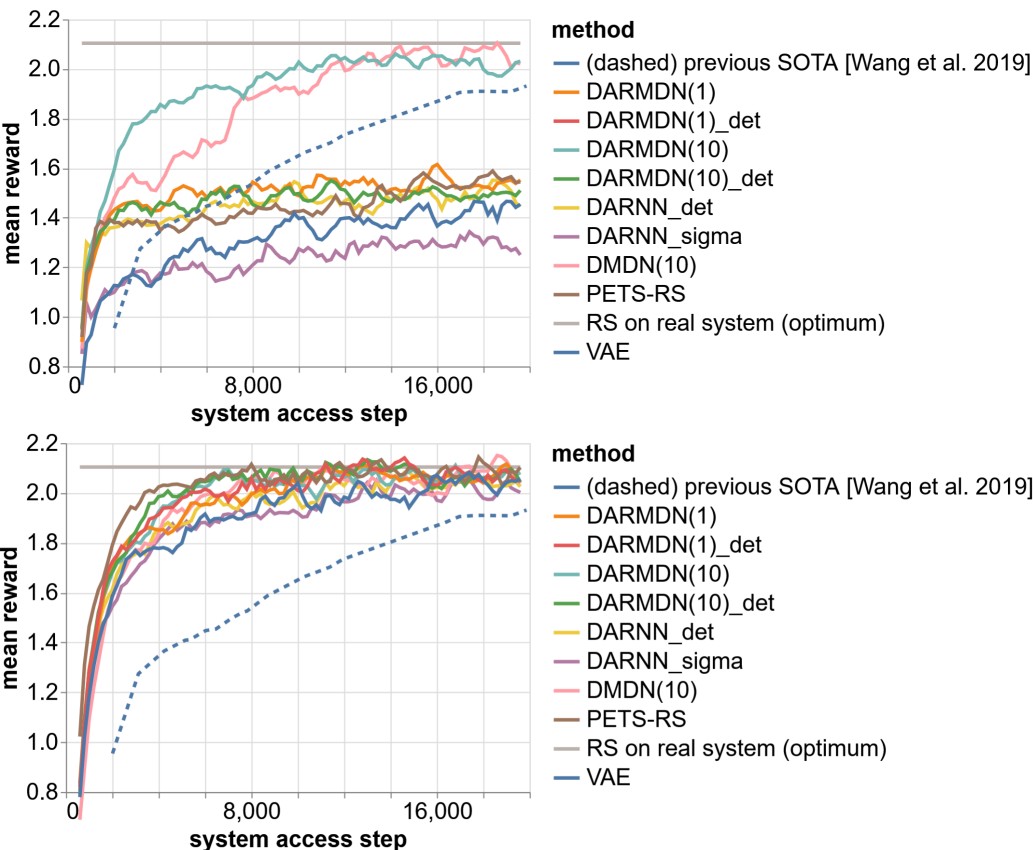

Figure 3: Acrobot learning curves on the raw angles (top) and sincos (bottom) systems. Reward is between 0 (hanging) and 4 (standing up). Episode length is $T = 200$, number of epochs is $N = 100$ with one episode per epoch. Mean reward curves are averaged across three to ten seeds and smoothed using a running average of five epochs, plotted at the middle of the smoothing window (so the first point is at step 600).

## F    THE POWER OF DARMDN: PREDICTING THROUGH CHAOS

Acrobot is a chaotic system (Ueda & Arai, 2008): small divergence in initial conditions may lead to large differences down the horizon. This behavior is especially accentuated when the acrobot slowly approaches the unstable standing position, hovers, "hesitates" which way to go, and "decides" to fall back left or right. Figures 4 and 5 depict this precise situation (from the test file of the "linear" data,

see Section 2.3): around step 18 both angular momenta are close to zero and $\theta_1 \approx \pi$. To make the modelling even harder, $\theta. = \pi$ is the exact point where the trajectory is non-continuous in the raw angles data, making it hard to model by predictive densities that cannot handle non-smooth traces.

In both figures we show the ground truth (red: past, black: future) and hundred simulated traces (orange) starting at step 18. There is no "correct" solution here since one can imagine several plausible "beliefs" learned using limited data. Yet it is rather indicative about their performance how the different models handle this situation.

First note how diverse the models are. On the sincos data (Figure 4) most posterior predictives after ten steps are unimodal. GP and DARMDN(10) are not, but while GP predicts a coin toss whether Acrobot falls left or right, DARMDN(10) bets more on the ground truth mode. Among the deterministic models, both DARNN$_{\text{det}}$ and DARMDN(10)$_{\text{det}}$ work well one step ahead (on average, according to their R2 score in Table 3), but ten steps ahead DARMDN(10)$_{\text{det}}$ is visibly better, illustrating its excellent R2(10) score.

On the raw angles data (Figure 5) we see a very different picture. The deterministic DARNN$_{\text{det}}$ picks one of the modes which happens to be the wrong one, generating a completely wrong trajectory. DARMDN(10)$_{\text{det}}$ predicts average of two extremem modes (around $\pi$ and $-\pi$), resulting in a non-physical prediction ($\theta_1$) which has in fact zero probability under the posterior predictive of DARMDN(10). The homoscedastic DARNN$_\sigma$ has a constant sigma which, in this situation is too small: it cannot "cover" the two modes, so the model picks one, again the wrong one. The heteroscedastic DARMND(1) correctly outputting a huge uncertainty, but since it is a single unimodal Gaussian, it generates a lot of non-physical predictions between and outside of the modes. This shows that heteroscedasticity without multimodality may be harmful in these kinds of systems. Finally, DARMDN(10) has a higher variance than on the sincos data, especially on the mode not validated by the ground truth, but it is the only model which puts high probability on the ground truth after ten steps, and whose uncertainty is what a human would judge reasonable.

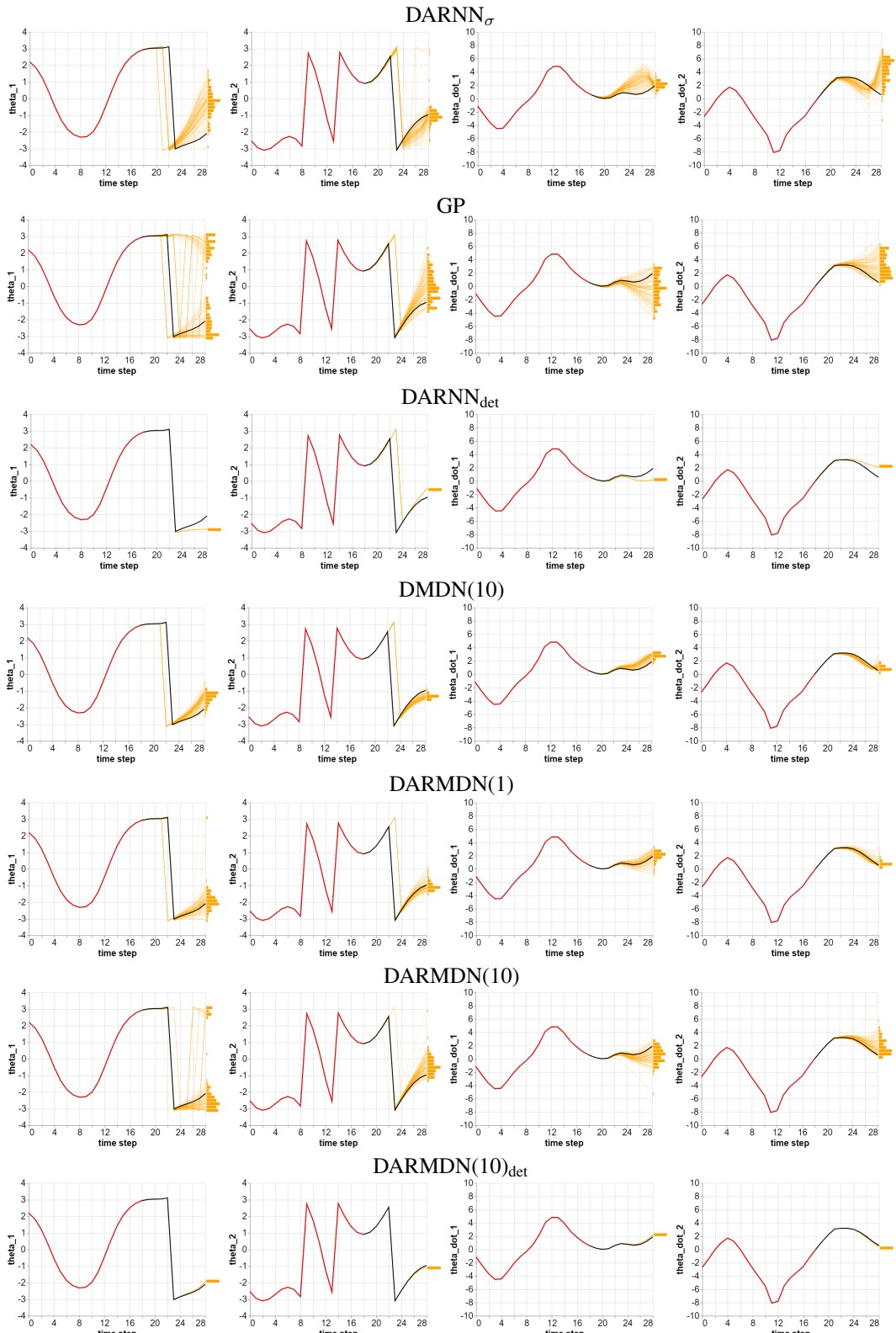

Figure 4: Ground truth and simulation of "futures" by the models trained on the sincos system. The thick curve is the ground truth, the red segment is past, the black segment is future. System models start generating futures from their posterior predictives at step 18. We show a sample of hundred trajectories and a histogram after ten time steps (orange).

.

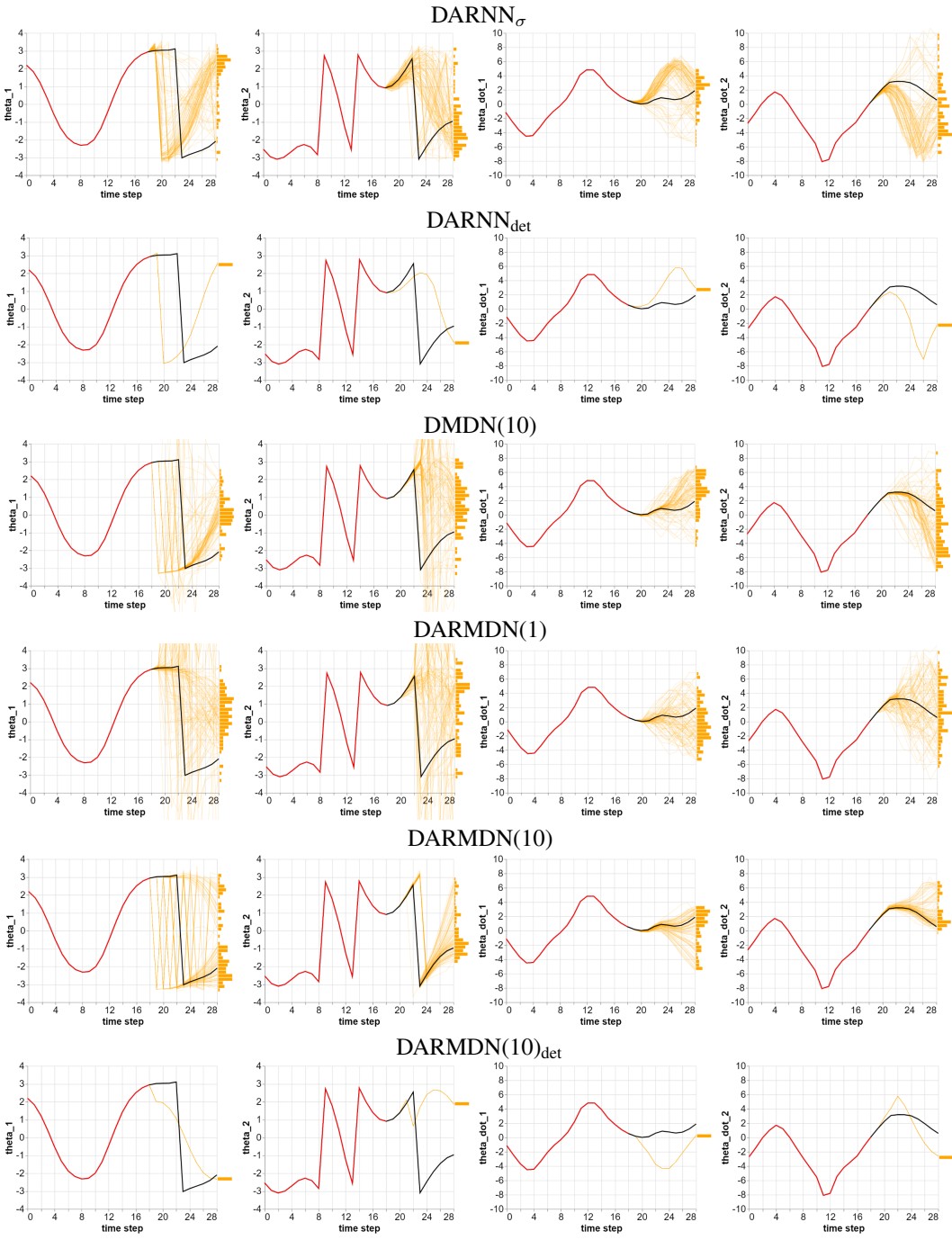

Figure 5: Ground truth and simulation of "futures" by the models trained on the raw angles system. The thick curve is the ground truth, the red segment is past, the black segment is future. System models start generating futures from their posterior predictives at step 18. We show a sample of hundred trajectories and a histogram after ten time steps (orange).

