# OpenReview forum: "Model-based micro-data reinforcement learning: what are the crucial model properties and which model to choose?"
_ICLR.cc/2021/Conference — ICLR 2021 Poster_

### Official Review · AnonReviewer4 · 2020-10-27
**Random shooting and generative models**

**Rating:** 7
**Confidence:** 3

**Review:**

This paper analyses the random shooting control strategy in combination with
various generative models \( p(\by_{t+1} | \mathcal T_t) \), which model
observations \(\by\) conditioned on the history of observation-action pairs.
The authors select two variants of the Acrobot environment to make
requirements like multimodal posteriors more explicit. Both statistics on the
distribution associated with the generative model under a fixed policy and
reward-dependent metrics were defined and analysed for a range of models.

The paper's contribution are twofold. First, the suggested experimental
protocol for model evaluation and benchmarking extends the usual evaluation
process in reinforcement learning, which often focuses purely on the
cumulative reward. The framework described introduces a range of ``static''
likelihood-based metrics. These static metrics are evaluated under a fixed
(potentially stochastic) policy and allow the separation of the model-based
control strategy and the underlying model for evaluation purposes.
Reward-based ``dynamic'' metrics are evaluated under a random-shooting
control mechanism. This framework heavily simplifies model evaluation by
providing evaluation metrics and fixing the environment and control strategy.
Under these assumptions this approach allows direct comparison and even
visualisation of various quantities of interest, including trajectories of
the one-step forward model. Second, this work applies the conceived framework
to evaluate a range of models on two different environments. These
environments are intended to make requirements like probabilistic posteriors
explicit. The authors conclude by claiming that 1. ``Probabilistic models are
needed when the system benefits from multimodal predictive uncertainty'', and
2. Deterministic models are sufficient if trained ``with a loss allowing
heteroscedasticity''.

There's an inherent trade-off between simplicity of the study and generality
its conclusion. While some of the simplifying assumptions made make this kind
of study possible in the first place, they also raise a range of questions:
- How appropriate are these metrics for problems with higher observation
  spaces? Can we expect the variance of estimates to increase and ratios and
  likelihoods to diminish by multiple orders of magnitude?
- Do the claims presented as ``important findings'' generalise to other
  environments?
- Does the correlation of the explained variance with the dynamic metrics
  hold on other environments?

A small note: a clear definition of micro-data reinforcement learning is
missing, and MBRL is introduced twice with conflicting definitions.

The future directions outlined by the authors of extending the results to
larger systems and other planning strategies are very relevant to reduce the
concerns of generalisability of the results and applications to problems of
higher dimensions. At the same time these modification will increase the
experimental complexity. This paper serves as a suitable baseline for
reference for future work to answer these questions. Therefore I consider
this paper a valid contribution to ICLR.

---

> ### Author Response · Authors · 2020-11-17
> **review answers**
>
> **The future directions outlined by the authors of extending the results to larger systems and other planning strategies are very relevant to reduce the concerns of generalisability of the results and applications to problems of higher dimensions. At the same time these modification will increase the experimental complexity.**
>
> We wholeheartedly agree on this, and we thank you for pointing this out. We refer to our general comments to all reviewers.
>
> **A small note: a clear definition of micro-data reinforcement learning is missing, and MBRL is introduced twice with conflicting definitions.**
>
> We added a paragraph on micro-data RL to the Introduction. We changed the wording in the abstract to "micro-data  model-based reinforcement learning (MBRL)".

---

### Official Review · AnonReviewer2 · 2020-10-28
**A nice experimental papers**

**Rating:** 7
**Confidence:** 2

**Review:**

##########################################################################

Summary:
This is a nice experimental paper on comparing various methods for MBRL. They listed properties that are desirable for MBRL, and defined multiple static metrics and dynamic metrics. They discussed popular methods within this framework and performed experiments in one task (two conditions) with the same control policy. Overall, the paper provides a good benchmark for MBRL.

##########################################################################

Reasons for score:
I think this paper provides a good benchmark and a good discussion and comparison between existing methods. This is important for future progress in MBRL. The code is promised to be publicly available, which I think it’s very crucial for an experimental paper. Bonus points if the authors can discuss how the results in this task might generalize to other different tasks (or actually performed some simple experiments). I think the current task and conditions do serve its purpose, but it just leaves the reader wondering about how the result might generalize.

##########################################################################

Pros:

1. The writing is very clear and easy to follow, also without redundancy. The paper is also well-structured. I particularly like that the authors clearly indicated which direction is better for the metrics in the tables.
2. The seven properties and metrics are mostly well-motivated and well-defined.
3. The authors discussed the results clearly with implications, not just stating which one is better, but also discussed when and why it is better.
4. The two important findings seem very interesting to me. I think the result of the necessity of probabilistic vs. deterministic models in different scenarios is a good contribution to this field.
5. The authors noted in the appendix that the code will be made public if accepted.

##########################################################################

Cons:
1. The authors provided motivation to the metrics they proposed. It would be great to see a brief description of how different metrics related to each other, what (different) aspects they capture, and whether/how they correspond to the seven desirable properties. Also, it seems not sufficient enough to filter out methods just because of the impossibility of computing the defined metrics.
2. Generalization to other tasks and different control policies. The authors did point this out as future work, but it’s also good to briefly discuss how it might or might not generalize.

#########################################################################

typo:
Introduction 5th paragraph: MBLR -> MBRL

---

> ### Author Response · Authors · 2020-11-17
> **review answers**
>
> **The authors provided motivation to the metrics they proposed. It would be great to see a brief description of how different metrics related to each other, what (different) aspects they capture, and whether/how they correspond to the seven desirable properties.**
>
> We refer here to our answer to AnonReviewer5 on point 13.
>
> **Also, it seems not sufficient enough to filter out methods just because of the impossibility of computing the defined metrics.**
>
> Yes, we agree, that's why Table 2 contains more techniques than Table 3. We included VAE and RealNVP in the dynamic tests. They are at a practical disadvantage in the sense that without some of the static metrics they are harder to debug and tune, but they are absolutely good contenders in the dynamic tests. Deterministic models also obviously lack some of the static metrics yet they perform very well on the environment that requires no multimodal posterior predictives.
>
> Second, in the library that we will publish soon, we came up with a scheme where all generative models can be turned into approximate mixture models that will allow the computation of Monte-Carlo LR, OR, R2 and KS.

---

### Official Review · AnonReviewer3 · 2020-10-28
**A well written paper that provides great insights of essential properties of good models for model-based RL**

**Rating:** 7
**Confidence:** 3

**Review:**

This paper tackles a key problem in model-based RL that is to identify the essential properties a predictive model needs to have to achieve good control performance. They solve this problem by systematically accessing the performance of a family of autoregressive mixtures learned by deep neural nets (DARMDN) using a fixed control strategy (i.e., random shooting). An important finding from this investigation is that deterministic models, when trained with a likelihood score in a generative setup, has comparable performance with the probabilistic modes. Furthermore, they proposed several static metrics (LR, OR, R2, KS, long-horizon R2, and long-horizon KS) which can be used to assess the performance of the models before having to run long experiments on the dynamic systems. These metrics can help to debug the models when evaluated dimension-wise. They found out that R2 with a 10-steps horizon correlates the most with dynamic performance. The findings lead to state-of-art sample complexity on the Acrobot system by applying an aggressive training schedule.

Strength:
1) The paper is well-written and structured.
2) The paper provides some good practices for evaluating model-based RL which will help to advance the field.
Weakness:
1) The paper only used one simple problem (Acrobot) and a random shooting control strategy to evaluate the different predictive models of the dynamic system, it'll be more interesting to see if the findings stills hold in more realistic and complex RL problems

-------------------------------------------------------------------------------------------------------------------------------------------------------------------------

Update:
After going through all the discussion, I'd be happy to raise my score to 7. The authors did a great job in clarifying all the concerns raised by the reviewers. This make the paper a much stronger publication.

---

> ### Author Response · Authors · 2020-11-17
> **review answers**
>
> **The paper only used one simple problem (Acrobot) and a random shooting control strategy to evaluate the different predictive models of the dynamic system, it'll be more interesting to see if the findings stills hold in more realistic and complex RL problems**
>
> We agree with the reviewer on this. For a more detailed answer, we refer to  the general comments to all reviewers and to our answer to AnonReviewer5 on points 3, 3a, and 9.

---

### Official Review · AnonReviewer1 · 2020-11-02

**Rating:** 5
**Confidence:** 3

**Review:**

The paper discusses the quality of surrogate models for model-based RL.  The two main contributions are first: defining the a set of evaluation metrics + experimental design and secondly performing a comparative analysis over existing methods.

There are several things I like about this paper:

- There is lack of existing work in throroughly comparing surrogate models in model-based RL
- The authors do a great job defining and arguing for evaluation metrics  (Section 2 is excellent)
- Defining requirements to the model that are falsifiable (2.1) are good to augment quantitative comparisons with qualitative ones

On the other hand there a several things I think that could be improved:

- Introduction:  The introduction  was quite confusing: Not only was it intermixed with conclusions that seem out of place. Also the discussion about uncertainty/probability was unclear.  Especially when talking about both uncertainty in a Bayesian sense (uncertainty over parameters or functions, epistemic) and "randomness" (stochastic effects of the environment) it is very easy
to be confusing.  E.g  "Probabilistic models are needed when the system benefits from multimodal predictive uncertainty."  It is not clear here what you mean by "probabilistic":  Parametric uncertainty (like in GPs) or models that can model multimodalities (like MDN)?

- Relevant work:  There is relevant work missing here.  Methods dealing with Bayesian deep learning[1,2], contextual auto-encoders[3].

   - [1] Gal, Yarin, Rowan McAllister, and Carl Edward Rasmussen. "Improving PILCO with Bayesian neural network dynamics models." Data-Efficient Machine Learning workshop, ICML. Vol. 4. 2016
   - [2] Depeweg, Stefan, et al. "Decomposition of uncertainty in Bayesian deep learning for efficient and risk-sensitive learning." International Conference on Machine Learning. PMLR, 2018.
   -[3] Moerland, Thomas M., Joost Broekens, and Catholijn M. Jonker. "Learning multimodal transition dynamics for model-based reinforcement learning." arXiv preprint arXiv:1705.00470 (2017).


- Limited scope: Only considering a single benchmark (in the sense of a single environment) is too limited. Chances are the "surprising results" are due to the specifics of this particular problem.  In general:   Methods that can model stochastic effects (e.g. neural networks that also predict a heteroscedastic Gaussian noise) should perform well when there is true stochasticity in the benchmark. Bayesian approaches (that model some form of parametric/functional uncertainty) should perform well when data is limited.

Overall I think this paper is written quite well,  deals with a topic that is relevant. While it is rigorous in terms of defining metrics and experimental set-up, it would greatly improve if additional benchmarks (and possibly methods) are considered.

---

> ### Author Response · Authors · 2020-11-17
> **review answers**
>
> **Introduction: The introduction was quite confusing: Not only was it intermixed with conclusions that seem out of place. Also the discussion about uncertainty/probability was unclear. Especially when talking about both uncertainty in a Bayesian sense (uncertainty over parameters or functions, epistemic) and "randomness" (stochastic effects of the environment) it is very easy to be confusing.**
>
> We do not want to take sides one way or another in the grand debate over how to *understand* the different types of uncertainties in learning systems, rather feed the debate with empirical evidence. Acrobot is a noiseless system, so our findings are definitely not about modelling stochasticity.
>
> **E.g "Probabilistic models are needed when the system benefits from multimodal predictive uncertainty." It is not clear here what you mean by "probabilistic": Parametric uncertainty (like in GPs) or models that can model multimodalities (like MDN)?**
>
> That is exactly the distinction we do not want to make, so we stay at a weaker position, talking about posterior predictives (in the Bayesian terminology), on which those approaches are directly empirically comparable. Probabilistic here means that more than a single prediction point has nonzero probability (as opposed to deterministic). Given this is what we would like to mean, we would be happy to reword to make it more understandable, if the reviewer has a suggestion. Perhaps the reviewer prefers "generative"?
>
> Are there any other places where our wording is confusing? It's our utmost goal to make our nonintuitive findings as understandable as possible.
>
> We added a (penultimate) paragraph to the introduction, attempting to clarify our stance.
>
> **Relevant work: There is relevant work missing here. Methods dealing with Bayesian deep learning[1,2], contextual auto-encoders[3].**
>
> Thank you for these references. We added [1,2] in the penultimate paragraph in the Introduction, and [3] in the experimental section when we introduce VAE.
>
> **Limited scope: Only considering a single benchmark (in the sense of a single environment) is too limited. Chances are the "surprising results" are due to the specifics of this particular problem. In general: Methods that can model stochastic effects (e.g. neural networks that also predict a heteroscedastic Gaussian noise) should perform well when there is true stochasticity in the benchmark. Bayesian approaches (that model some form of parametric/functional uncertainty) should perform well when data is limited.**
>
> We are very happy that the reviewer explicitly states this commonly held belief. Acrobot is a noiseless system, so our finding is a direct counterexample to "neural networks that also predict a heteroscedastic Gaussian noise should perform well when there is true stochasticity in the benchmark", which is exactly why we think it is worthy to publish. While we have no explanation (only hypotheses) why heteroscedasticity helps when all uncertainties are epistemic, the empirical observation is solid, reproducible, and that should be a valid paper (like it would be in all experimental sciences).
>
> It is definitely our next research step to see if the findings apply to other systems, but observing a single counterexample to a widely held belief is an interesting result even in itself.
>
> **While it is rigorous in terms of defining metrics and experimental set-up, it would greatly improve if additional benchmarks (and possibly methods) are considered.**
>
> For "additional benchmarks", we agree completely. We refer here to the general comments to all reviewers.
>
> We are not planning to include additional methods, for the lack of man- and computational power. On the other hand, we will open the code and document the model API so experts of other techniques can easily add their modelling techniques to the pool.

---

### Official Review · AnonReviewer5 · 2020-11-06
**Interesting methods proposed to evaluate dynamics models, but confusing paper and not enough evaluation**

**Rating:** 6
**Confidence:** 4

**Review:**

Paper Summary:

This paper performs a detailed ablation study over different mechanisms for predicting dynamics for model-based control. The paper proposes its own metrics for models, evaluates how different types of uncertainty impact predictions, and measures control performance with random shooting MPC. By implementing a new hyper parameter schedule, the paper shows new SOTA performance (in terms of sample efficiency) on the acrobat task.

-----

Rebuttal phase:
I am updating my score based on the numerous clarifications of experiments and improvements of framing the paper.
-----


Score Summary:

The analysis in this paper is very warranted, but the methodology executed raises many questions as to if the claims will be generalizable. The paper evaluates models on only one task and on their own metrics, so there is difficulty in matching the paper to the literature. This difficulty of placement for a paper making broad claims, leads to a currently unpublishable form.



-----


Comments:
0) This type of work is absolutely critical to advancing MBRL. I have a lot of criticisms below, but I want to say that I think this could be a very good paper when all is said and done. The metrics proposed are relevant and the breadth of models tested is encouraging.

1) the authors refer to micro-data in the title and multiple times in the introduction, but never define this. What regime of data do the authors consider micro data? Is it a cap at 1000 points or depend on the environment?
1b) Where did the number of training points at 5000 come from in 2.3?

2)The authors make some strange claims in the paper, that vary from interesting to unsupported and too strong. Many of these claims need to be backed by citations.
2a) "Unlike computers, physical systems do not get faster with time" is playing loose and fast with the fact that the computational power of computers is increasing rapidly in recent years. This does not really have much of a baring on the rest of the paper and should be tied in better.

3) The evaluation of only fixed, random-shooting MPC is okay, but limited. For a paper that is titled "what are the critical model properties to choose" I would like to see more acknowledgement that the model one chooses is heavily dependent on the method it is being used for.
3a) PETS (Chua et. al 2018) as a paper is cited, can the authors comment why the trajectory propagation method or an informed sampling like CEM is not considered? For example, in appendix A.8, the PETS paper shows performance is dramatically better with CEM, and other methods such as POPLIN (Wang et. al 2019) and PDDM (Nagabandi et. Al 2019) have built on this, it seems like a step back to not consider these advances.

4) The model hyper-parameters section is hard to read (A.D) and is crucial to being able to trust the results. What search method was performed over the ranges given per model? And, how extensive was said search?
4b) The authors claim "system modeling for MBRL is essentially a supervised learning problem with AutoML." To my knowledge there are no detailed examples of MBRL using AutoML?

5) the paper claims SOTA sample efficiency on acrobat, but I am not sure. Can the authors provide more information. For example, I am looking at a MBRL benchmarking paper and the reward takes on a vastly different range (https://arxiv.org/pdf/1907.02057v1.pdf). Acrobat is not the most well studied environment in recent years, so this discrepancy is hard to reason with.

6) While the model requirements and evaluation metrics are insightful, it could be useful to distill them into slightly more compact analysis. What are the most important takeaways for each is interesting, but more useful is "which type of model does each person want." It seems unlikely that every problem should use the same model.

7) The number of action samples used in the random shooting is extremely low at n=100 sequences. Was this validated against? It seems at least an order of magnitude lower than many other papers in the area.

8) In figure 1, the paper shows "how different model types deal with uncertainty" which types of models were actually used, or is this an illustration?

9) The claim that random-shooting on the real environment is optimal control on the real environment is very troubling (especially with the number of action samples). Random shooting is only guaranteed to be optimal when the number of samples goes to infinity. If this metric was meant as "matches the state of the art for control of the acrobat" this should be clarified.

10) A more complicated environment is needed to make many of these claims more believable. Were any experiments done on hopper or half cheetah?

11) What does "starting from an approximately hanging position" mean for acrobat, and is that the standard used in other control papers? Please classify this numerically.

12) the claim that model accuracy is the bottleneck of MBRL should be cited in the introduction. I think there is some preliminary work showing that direction, but sampling based control is also sub-optimal.

13) Are certain requirements correlated with certain metrics? This would be an interesting analysis, but there wasn't much of it in the paper (yes low on space, I know :/)

14) was the likelihood bound of 1.47E-6 taken from somewhere or how was this number computed?

15) The models and results section was confusing. I think the reader would benefit from a discussion of the results. There are interesting findings, such as what the authors discussed on training with uncertainty, that are not always super clear.

-----

Minor Points:
- Paragraph 2 of second page, "MBLR"
- inconsistency in hyphenating one-step "one step ahead"
- "Robust and computationally efficient probabilistic generative models are the crux of many machine" needs some commas or punctuation.
- Fragment section 2 "So our goal is to learn p()"
- Section 3 starts with "A commonly held belief," just say what the results show.
- Footnote 2, I agree it is annoying at times, but most papers report the trial length, no?
- The use of our in the introduction is slightly strange
- Generally accepted use fo quotes at the end of a sentence is "hi."
- The analogy to Yann Lecun is strange and un-cited.

---

> ### Author Response · Authors · 2020-11-17
> **Points 10-15 and minor points**
>
> **10 A more complicated environment is needed to make many of these claims more believable. Were any experiments done on hopper or half cheetah?**
>
> We refer here to the general comments to all reviewers.
>
> We would also like to emphasize the difference between "believability" and "generalizability". It is possible that Acrobot is an exceptional system, so our results don't generalize to other systems (in which case we definitely need to understand what makes Acrobot special), but our results are fully reproducible.
>
> We added a disclaimer in the introduction (P2 on page2) to allow for the possibility that the results do not generalize to other systems.
>
> **11 What does "starting from an approximately hanging position" mean for acrobat, and is that the standard used in other control papers? Please classify this numerically.**
>
> This is the starting position obtained with the default reset function of the acrobot open AI gym environment: all state variables (two angles, and velocities) are uniformly sampled in [-0.1, 0.1]. This is the standard for papers using acrobot as a benchmark environment (see e.g Wang et al. 2019). We added details in the paper
>
> **12 the claim that model accuracy is the bottleneck of MBRL should be cited in the introduction. I think there is some preliminary work showing that direction, but sampling based control is also sub-optimal.**
>
> We added references that emphasize the importance of model performance and softened the claim to "one of the important bottlenecks".
>
> **13 Are certain requirements correlated with certain metrics? This would be an interesting analysis, but there wasn't much of it in the paper (yes low on space, I know :/)**
>
> There is no direct mapping, but we defnitely had robustness and debuggability (R7) in mind when we designed the metrics. We added a paragraph at the end of 2.2.1 to elaborate on how the metrics can be used in practice and what their relationships are.
>
> **14 was the likelihood bound of 1.47E-6 taken from somewhere or how was this number computed?**
>
> It's the value of the 1D Gaussian density at 5 standard deviations, as a salute to 5 sigma in experimental physics as the threshold of discovery. It is more metaphoric than formal but for the purposes of this paper it worked will (OR is very low but nonzero, indicating the model mismatch, relative between the different models).
>
> **15 The models and results section was confusing. I think the reader would benefit from a discussion of the results. There are interesting findings, such as what the authors discussed on training with uncertainty, that are not always super clear.**
>
> Without more specific comments it is hard for us to understand what the Reviewer would like us to change, and we don't want to guess. We would love to obey here since it is also our interest to make this section as clear as possible. Would it be possible to give us more details here?
>
> **inconsistency in hyphenating one-step "one step ahead**
>
> The rule is "one-step" when used as an adjective (as in "one-step optimality"), and "one step" when used as a noun (as in "one step ahead").
>
> **Section 3 starts with "A commonly held belief," just say what the results show.**
>
> This paragraph motivates the *models* part of the section. It's too specific to put in the Introduction section. We also think that a lot of readers don't read papers from the beginning to the end, so having standalone sections is useful.
>
> **Footnote 2, I agree it is annoying at times, but most papers report the trial length, no?**
>
> Yes, it is as if we had to know the test set size in supervised learning (different in every paper), because authors report the number of test errors, not the rate. It's also why you can't compare directly our curves to [Wang et al. 2019] or others.
>
> **The use of our in the introduction is slightly strange**
>
> We are not sure what you mean here.
>
> **The analogy to Yann Lecun is strange and un-cited.**
>
> We agree that it is unusual since self-supervised learning is usually used in computer vision ot NLP, but we want to push the argument that system logs (time series) are also an important instantiation of the paradigm. There was a colored link in the paper; we explicitly give it in a footnote now. We also know that linking a facebook discussion may also be seen as unusual, but note that this is not a formal claim that needs to be peer reviewed, rather an opinion that gives an additional motivation to the research direction. The discussion at the link is important and rich.

---

> > ### Comment · AnonReviewer5 · 2020-11-17
> > **Thank you for detailed responses**
> >
> > I just wanted to thank the authors for their thorough and good-faith responses to every point here. I will be going through again in detail later today / tomorrow, so hopefully we can have a constructive discussion period.

---

> > ### Comment · AnonReviewer5 · 2020-11-17
> > **responses**
> >
> > Thanks for taking the time on all these small points.
> >
> > Most of the important things are above, and just wanted to thank you for helping me improve my grammar
> > "The rule is "one-step" when used as an adjective (as in "one-step optimality"), and "one step" when used as a noun (as in "one step ahead")."
> >
> > I'm curious to hear on some more discussion of the introduction, but the authors have done a lot of work to improve this paper and I hope the other reviewers read some of this discussion.

---

> > > ### Author Response · Authors · 2020-11-21
> > > **Thank you for the discussion**
> > >
> > > We thank you again and greatly appreciate the time you put into this review and the ongoing discussion.

---

> ### Author Response · Authors · 2020-11-17
> **Points 5-9**
>
> **5 the paper claims SOTA sample efficiency on acrobat, but I am not sure. Can the authors provide more information. For example, I am looking at a MBRL benchmarking paper and the reward takes on a vastly different range (https://arxiv.org/pdf/1907.02057v1.pdf). Acrobat is not the most well studied environment in recent years, so this discrepancy is hard to reason with.**
>
> To compare directly with the previous SOTA, we added the PETS-CEM curve from the paper the reviewer cites (Wang et al. 2019) onto our graph. Their reward is the return obtained on a 200-step episode. The conversion is our_mean_reward = their_reward / 200 + 2.
>
> **6 While the model requirements and evaluation metrics are insightful, it could be useful to distill them into slightly more compact analysis. What are the most important takeaways for each is interesting, but more useful is "which type of model does each person want." It seems unlikely that every problem should use the same model.**
>
> We fully agree with the reviewer on this point, but we simply could not get into this more deeply in this paper. For this, one would need to sweep a larger set of problems and application scenarios. We believe that we give enough help on the requirements so that a practitioner can prioritize them, then go to Table 1 (or generate other lines for other methods) to choose the methods or families to consider.
>
> In our situation (we deliver tunable models to systems engineers), R3, R6, and R7 have definitely higher priority than in a research paper, and we believe this is a common scenario in industrial settings.
>
> **7 The number of action samples used in the random shooting is extremely low at n=100 sequences. Was this validated against? It seems at least an order of magnitude lower than many other papers in the area.**
>
> See answer to 3a).
>
> **8 In figure 1, the paper shows "how different model types deal with uncertainty" which types of models were actually used, or is this an illustration?**
>
> Yes,  this is essentially an illustration, but you are right that we should add the model types from Fig4-5, and we did.
>
> **9. The claim that random-shooting on the real environment is optimal control on the real environment is very troubling (especially with the number of action samples). Random shooting is only guaranteed to be optimal when the number of samples goes to infinity. If this metric was meant as "matches the state of the art for control of the acrobat" this should be clarified.**
>
> Yes, we agree and we corrected this misunderstandable claim. We definitely do not mean optimality in a theoretical sense, rather in the sense of the last sentence of the reviewer. We now write: "Second, Acrobot is one of the systems where i) random shooting applied on the real dynamics is the state of the art in an experimental sense and ii) random shooting combined with good models is the best approach among MBRL (and even model-free) techniques \citep{Wang2019}."

---

> > ### Comment · AnonReviewer5 · 2020-11-17
> > **response**
> >
> > These clarifications are well done.

---

> ### Author Response · Authors · 2020-11-17
> **Points 3-4**
>
> **3. The evaluation of only fixed, random-shooting MPC is okay, but limited. For a paper that is titled "what are the critical model properties to choose" I would like to see more acknowledgement that the model one chooses is heavily dependent on the method it is being used for.**
>
> It is indeed true that a model can be built or chosen depending on the method or agent one wants to use. Maybe the reviewer has in mind solutions such as MuZero (Schrittwieser et al., 2019) where one does not need a model of the underlying dynamics. The comparison done in the paper is limited to models of a system dynamics. We believe, without being able to guarantee it, that the conclusion on the different model properties would still hold with different methods (e.g Dyna style) if one only considers models of a system dynamics.
> Of course if the reviewer provides more details defending this point, we are willing to acknowledge it in the paper.
>
>
> **3a) PETS (Chua et. al 2018) as a paper is cited, can the authors comment why the trajectory propagation method or an informed sampling like CEM is not considered? For example, in appendix A.8, the PETS paper shows performance is dramatically better with CEM, and other methods such as POPLIN (Wang et. al 2019) and PDDM (Nagabandi et. Al 2019) have built on this, it seems like a step back to not consider these advances.**
>
> We fully agree with the reviewer, and we included a disclaimer in Section 2.2.2. and details in Appendix D. We agree that we might need better planners or full-blown model-free RL (learned on the model) to beat the SOTA on more complex systems. However we judged RS with n=100 to be sufficient for our study. We selected this planning strategy versus RS with a larger n or CEM by evaluating their performance on the true dynamics.  Although n=500 (for both RS and CEM) leads to a better performance (average reward around 2.3 instead of 2.1), n=100 was already sufficient to achieve more than decent rewards (average reward > 2, beating the SOTA result presented in the benchmark paper of Wang et al. 2019) while reducing the total computational cost of the study. We also note that using n=1000 leads to a performance very similar to using n=500 indicating that considering a larger number is not relevant here.
>
> Finally, we want to stay within a scientific paradigm where the goal is to understand elements of the system (rather than in an engineering paradigm where the goal is to optimize the solution by any means possible). In fact, you could lay the same criticism on the PETS paper that they did not explore the different properties of the models (and learning schedule) and focused too much effort on optimizing the planner.
>
> **4. The model hyper-parameters section is hard to read (A.D) and is crucial to being able to trust the results. What search method was performed over the ranges given per model? And, how extensive was said search?**
>
> We did a two-step random search, first on a broader, then on a finer grid. We added a paragraph to Appendix D. Also note that the published code will contain not only the optimal (tuned) models but also the hyperparameter optimization code. It is there not only for complete reproducibility, but also for easy reusability of the models on other systems (which is our ultimate goal).
>
> **4b) The authors claim "system modeling for MBRL is essentially a supervised learning problem with AutoML." To my knowledge there are no detailed examples of MBRL using AutoML?**
>
> What we mean here is what we write right after: models need to be retrained and if needed, retuned often, on data sets of different distribution whose size may vary by orders of magnitude, with little human supervision. This does not mean we need to do full hyperopt in every episode (step 2 in 2.2.2), rather that the model we choose should be robust, trainable without human babysitting over a range of different distributions and data sizes. One catastrophic learning failure (e.g. getting stuck in initial random function) means the full MBRL loop goes off the rail. Models that _need_ to be retuned (because of sensitivity to hyperparameters) must have the retuning (AutoML) feature encapsulated into their training. The models ended up on the top were not sensitive to the choice of hyperparameters, so we did not need to retune them in every iteration.
>
> Since this is a crucial issue and can be misunderstood, we added this explanation in 2.2.2. and changed the wording in the introduction.

---

> > ### Comment · AnonReviewer5 · 2020-11-17
> > **responses**
> >
> > ### 3
> >
> > With the above comments on specificity of analysis, I am more okay with just the random-shooting. Can't have everything in the page limit. For example, using  neural networks in real-time systems scales much better than gaussian processes with developments of embedded neural hardware. GPs scale with n^3 at predictions, and NNs can be nicely parallelized. Figure A.11 from the arxiv PETS paper is what I was thinking about https://arxiv.org/pdf/1805.12114.pdf. With my comment above about other groups using learned-models for control, that's what I was thinking of model-task pairing.
> >
> > ### 3a
> >
> > I openly say I took I had a biased framing of the paper, so I spent some time digging into details that are not the important points. As you say "Finally, we want to stay within a scientific paradigm where the goal is to understand elements of the system (rather than in an engineering paradigm where the goal is to optimize the solution by any means possible). In fact, you could lay the same criticism on the PETS paper that they did not explore the different properties of the models (and learning schedule) and focused too much effort on optimizing the planner." Improving the introduction and abstract to make sure the readers understand what the paper is trying to do is very helpful (as is being done).
> >
> > ### 4
> >
> > Great! That's a good search, that helps a lot for a paper like this. *_ will the environment data be released for a evaluation set? Some people don't have access to Mujoco_*. This is above the requirement by far, but is a good gesture.
> >
> > ### 4b
> >
> > Great. That makes sense. Something I don't understand why it is done, is in the PETS paper they train cheetah models incrementally and other models from scratch every time. This sort of obscurity is what I think your point is getting at.  AutoML for peak performance was a little different than avoiding a "really bad model" in my eyes.

---

> > > ### Author Response · Authors · 2020-11-21
> > > **Points 3, 3a and 4.**
> > >
> > > **3. For example, using neural networks in real-time systems scales much better than gaussian processes with developments of embedded neural hardware. GPs scale with n^3 at predictions, and NNs can be nicely parallelized. Figure A.11 from the arxiv PETS paper is what I was thinking about https://arxiv.org/pdf/1805.12114.pdf. With my comment above about other groups using learned-models for control, that's what I was thinking of model-task pairing.**
> > >
> > > We are mostly confronted with systems where the control frequency is not very large and not the bottleneck for now. However we agree that this is an important question and models with fast prediction times might be required depending on the applications. We added a sentence acknowledging this in the paper: "We note that depending on the application and the desired control frequency of the system, one may also require models with fast prediction times."
> > >
> > > **3a As you say "Finally, we want to stay within a scientific paradigm where the goal is to understand elements of the system (rather than in an engineering paradigm where the goal is to optimize the solution by any means possible). In fact, you could lay the same criticism on the PETS paper that they did not explore the different properties of the models (and learning schedule) and focused too much effort on optimizing the planner." Improving the introduction and abstract to make sure the readers understand what the paper is trying to do is very helpful (as is being done).**
> > >
> > > We think this is being addressed with our replies to points 1b and 2 above.
> > >
> > > **4 will the environment data be released for a evaluation set? Some people don't have access to Mujoco.**
> > >
> > > The environment does not require Mujoco so anyone will be able to generate the data. We might also release the datasets with the code depending on their size.

---

> ### Author Response · Authors · 2020-11-17
> **Points 1-2**
>
> **1. the authors refer to micro-data in the title and multiple times in the introduction, but never define this. What regime of data do the authors consider micro data? Is it a cap at 1000 points or depend on the environment?**
>
> We borrowed the term from robotics research (Chatzilygeroudis et al. 2020 and his thesis; Mouret 2016). In robotics they mean "approaches that tackle the challenge of learning by trial-and-error in a few minutes on physical robots". In Mouret 2016: "Any learning process that involves physical tests or precise simulations (e.g., computational fluid dynamics) comes up against the same issue. In short, while data might be abundant in the virtual world, it is often a scarce resource in the physical world. I refer to this challenge as 'micro-data' learning".
>
> We use it in a more general sense to say that the bottleneck of the task is data (interactions with the system) and that sample complexity (MRCP) is as important as the asymptotic performance (RMAR). The regime definitely depends on the environment, but it's essentially 10s to 1000s of interactions, not millions or billions.
>
> We added a full paragraph to the introduction and also elaborate the relationship of MBRL, micro-data learning, and engineering systems, in the third paragraph of the Introduction.
>
> **1b) Where did the number of training points at 5000 come from in 2.3?**
>
> We knew from literature (e.g., Wang et al. 2019) that this is the range where a new SOTA will need to perform well on Acrobot. The typical episode size on Acrobot is 200, we went slightly higher in the static sets (500) for reliable model selection and tuning, and we used 10-fold CV for variance control. With careful tuning the data size could probably be reduced, but we did not need to go there for the purposes of the paper and the longer term research agenda.
>
> **2. Unlike computers, physical systems do not get faster with time" is playing loose and fast with the fact that the computational power of computers is increasing rapidly in recent years. This does not really have much of a baring on the rest of the paper and should be tied in better.**
>
> We are a bit puzzled by what the reviewer means here. First, we almost literally quoted Chatzilygeroudis et al. 2020, a paper that we like very much (even though they work in a quite different domain from us). The quote is directly relevant to our applied projects where access to real systems is tightly controlled by systems engineers, and data collection is the major cost of the projects. Typically, 100s to 1000s of time steps are collected from which we can jump start the learning process (MBRL loop in Section 2.2.2.).
>
> If the reviewer's objection is with regard to the "unlike computers" phrase, our point here is that in the more common setup where RL is applied to simulators, data is not the bottleneck, and the research paradigm is that algorithmic development goes hand in hand with the steady increase of compute. In our case this is not an option, so even if the benchmark systems are simulated, algorithms should be developed to handle limited data.
>
> We extended the first three paragraphs of the Introduction to be more clear about how engineering systems call for both micro-data learning and model-based RL. We are not sure whether this answers to the reviewer's concern, and we are genuinely interested in discussing this further to improve the chain of arguments that motivate our research agenda.

---

> > ### Comment · AnonReviewer5 · 2020-11-17
> > **Responses**
> >
> > ### 1
> >
> > I think mostly I was a little surprised to not have heard of this before. I think the addition of introduction material fits well.
> >
> > ### 1b
> >
> > That's very reasonable, thanks.
> >
> > I think where this paper could go from borderline to great is the level of analysis done for what is particular to Acrobat and what is generalizable. The statement you made in the general review about vertical vs horizontal papers is very good. It could be worth saying something in the intro akin to "this paper uses a very specific study to make specific comparisons" (that you detail).
> >
> > ### 2
> > I am still not entirely sold on the framing, but I think it is partially because the current state of MBRL in confusing. I am also starting to understand it better.  Can the insights the paper shows about model-learning also be useful for other areas of control than MBRL (ie control theory, where model learning isn't iterative, but is used at control time similarly) ?  Like what is done with Acrobot -- focusing the paper on one analysis is okay, because it gives more room to discuss things, but acknowledge this as an angle people could take.
> >
> > This is part my case -- for most systems, we know *something* about the system a-priori, so we should be able to have priors / some simulator to use to create an original model estimate. That model estimate can be very useful in the micro-data regime. Some works are moving towards model-based value initialization (Mo Chen et. al 2020) and using simplified models to estimate dynamics.
> >
> > Is the goal in this case, Model-based RL that could work in any robot in a very short period of time? 5000 datapoints at 50hz is less than 2 minutes of data.

---

> > > ### Author Response · Authors · 2020-11-21
> > > **Points 1b and 2**
> > >
> > > **1b. I think where this paper could go from borderline to great is the level of analysis done for what is particular to Acrobat and what is generalizable. The statement you made in the general review about vertical vs horizontal papers is very good. It could be worth saying something in the intro akin to "this paper uses a very specific study to make specific comparisons" (that you detail).**
> > >
> > > This was the goal of paragraph 2 and 3 (page 2) of the introduction. We brought small changes to paragraph 2 and 3 to make it more clear.
> > > Paragraph 2: "We propose to run a comparison of popular probabilistic models on the Acrobot system **to** study the model properties required to achieve state-of-the art performances. We believed such ablation studies are missing from existing **horizontal** benchmark papers (Wang et al. 2019).
> > > Paragraph 3: we added a third reason (see answer to 2 below)
> > >
> > > **2. Can the insights the paper shows about model-learning also be useful for other areas of control than MBRL (ie control theory, where model learning isn't iterative, but is used at control time similarly) ? Like what is done with Acrobot -- focusing the paper on one analysis is okay, because it gives more room to discuss things, but acknowledge this as an angle people could take.**
> > >
> > > We added a third reason in paragraph 3 to acknowledge this:
> > > "Third, using a single system allows both a deeper and simpler investigation of what might explain the success of popular methods."

---

### Author Response · Authors · 2020-11-17
**General comments to all reviewers.**

We are deeply grateful to all of you for your mindful, meaningful, and heartful reviews. Whatever the fate of the paper, it is these kinds of discussions that make the field move ahead. It was very touching to us to see how deep you dug into the topic and how much effort you put into your reviews. Your questions and remarks were quite challenging and sent us into a week of exciting internal discussions and experiments which, at the end, we believe, resulted in a much improved paper.


All the reviewers asked about the generalization of our results to other environments. We provide a common answer here.

First, we believe that a lot can be learnt from an in-depth study on a given environment. "Selective Dyna-style Planning Under Limited Model Capacity" (Zaheer et al. 2020) is another example of a very interesting paper recently published which only uses the Acrobot environment. Acrobot is a simple environment in some ways (low dimensional) but not so simple in others (linear models and homoscedastic NNs are clearly suboptimal unlike on, e.g., inverted pendulum that a linear model can solve very quickly). Furthermore it is always possible to tweak the environment to study different properties as we did with the raw angles vs sin-cos setup. We consider Acrobot similar to MNIST in the computer vision community. It is simple enough to run a lot of experiments to make and validate generalizable scientific statements. If we would like to follow a similar path in MBRL, we need to start somewhere, and Acrobot seems the right sized stepping stone.

Second, some of our findings are not completely surprising: our claim on the performance of deterministic models is supported in papers using different environments. For instance, quoting Appendix B.3 of the model-based RL benchmark paper by Wang et al. (2019) "PE-E is among the best models, with comparable performance to other combinations such as PE-DS, PE-TS1, PE-TSinf" (where PE refers to Probabilistic Ensembles and E to predicting the mean). Figure 4 of (Chua et al. 2018) clearly shows that deterministic models always perform as well as trajectory sampling strategies, except on Halfcheetah. As we wrote in one of our answers, it is possible that Acrobot is an exceptional system, so our results do not generalize to other systems (in which case we definitely need to understand what makes Acrobot special), but our results are fully reproducible.

Finally, as pointed out by one of the reviewers, the depth of this study was made possible because of some restrictive choices and there was indeed a trade-off between simplicity and generalizability. "Horizontal" papers like Wang et al, 2019 are also immensely useful but they rarely give insight to *why* certain methods work better or worse on certain problems. We agree that some questions still hold but these questions were in part raised *as a  result of this study* and will naturally require future work. We believe that this paper is a nice contribution to the community which will be able to refer to and compare to the presented results.

---

### Decision · Program_Chairs · 2021-01-07
**Final Decision**

**Decision:**

Accept (Poster)

**Comment:**

**Overview** This paper performs detailed ablation studies over different dynamics prediction methods for MBRL. It proposes metrics for models to evaluate how different types of uncertainty impact predictions. The paper also measures control performance with random shooting MPC. The paper further implements a new hyper parameter schedule to achieve new SOTA performance on the acrobat task.

**Pro**
- The paper is well-written.
- The analysis in this paper is very warranted.
- The paper provides a very detailed ablation study.
- The authors do a great job defining and arguing for evaluation metrics.
- The seven properties and metrics are mostly well-motivated and well-defined.
- The authors discussed the results clearly with implications.
- The result of the necessity of probabilistic vs. deterministic models in different scenarios is a good contribution to this field.

**Con**
- The methodology might be hard generalizable, i.e., there is difficulty in matching the paper to the literature based on its own defined metric.
- The scope might be limited.

**Recommendation** The paper provides a significant contribution to MBRL by providing a detailed empirical study. During the rebuttal phase, the authors addresses many reviewers' concerns in a satisfactory way.  The paper is well-written and easy to read. The recommendation is an accept.